# Neoadjuvant sintilimab in combination with concurrent chemoradiotherapy for locally advanced gastric or gastroesophageal junction adenocarcinoma: a single-arm phase 2 trial

In this multicenter, single-arm phase 2 trial (ChiCTR1900024428), patients with locally advanced gastric/gastroesophageal junction cancers receive one cycle of sintilimab (anti-PD1) and chemotherapy (S-1 and nab-paclitaxel), followed by 5 weeks of concurrent chemoradiotherapy and sintilimab, and another cycle of sintilimab and chemotherapy thereafter. Surgery is preferably scheduled within one to three weeks, and three cycles of adjuvant sintilimab and chemotherapy are administrated. The primary endpoint is the pathological complete response. Our results meet the pre-specified primary endpoint. Thirteen of 34 (38.2%) enrolled patients achieve pathological complete response (95% CI: 22.2-56.4). The secondary objectives include disease-free survival (DFS), major pathological response, R0 resection rate, overall survival (OS), event-free survival (EFS), and safety profile. The median DFS and EFS were 17.0 (95%CI: 11.1-20.9) and 21.1 (95%CI: 14.7-26.1) months, respectively, while the median OS was not reached, and the 1-year OS rate was 92.6% (95%CI: 50.1-99.5%). Seventeen patients (50.0%) have grade ≥3 adverse events during preoperative therapy. In prespecified exploratory biomarker analysis, CD3$^+$ T cells, CD56$^+$ NK cells, and the M1/M1 + M2-like macrophage infiltration at baseline are associated with pathological complete response. Here, we show the promising efficacy and manageable safety profile of sintilimab in combination with concurrent chemoradiotherapy for the perioperative treatment of locally advanced gastric/gastroesophageal junction adenocarcinoma.

Gastric and gastroesophageal junction (G/GEJ) cancers represent the fifth most common newly diagnosed malignancy and the fourth leading cause of cancer-related death worldwide[1]. In China, approximately 400,000 patients were diagnosed with G/GEJ cancers, and 289,000 individuals died of G/GEJ cancers in 2016[2]. Furthermore, nearly half of the gastric cancers in China are stage III or IV at diagnosis[3]. Despite

recent advances in multidisciplinary or multimodal therapies, the prognosis of locally advanced patients remains poor, with a median overall survival (OS) of only 34.4 months and a 5-year OS rate of only 38.7%[4].

According to the guidelines of the Chinese Society of Clinical Oncology (CSCO), a combination of perioperative therapy and D2

✉e-mail: guan-wx@163.com; baoruiliu@nju.edu.cn

gastrectomy is currently considered the standard treatment option for locally advanced G/GEJ cancers ($cT_{3}N_{2\text{-}3}M_{0}$ and $cT_{4a}N_{+}M_{0}$, or $cT_{4b}N_{any}M_{0}$ after multidisciplinary discussion)[5]. Neoadjuvant chemotherapy based on fluorouracil, platinum, taxanes, or chemoradiotherapy followed by surgery and adjuvant chemotherapy is recommended by National Comprehensive Cancer Network (NCCN) and the CSCO guidelines[5,6]. Nevertheless, the optimal perioperative therapeutic protocol and sequence remain undefined. Although perioperative therapy can lead to a high R0 resection rate of more than 90%[7–9]. Therefore, the optimization of perioperative treatment regimens is warranted to improve the clinical outcomes of patients with locally advanced G/GEJ cancers.

## Table 1 | Baseline patient characteristics

| Characteristic | Total (n = 34) |
| --- | --- |
| Age, years, median (range) | 65.5 (58–68) |
| Male, n (%) | 28 (82.4) |
| ECOG PS score of 0, n (%) | 26 (76.5) |
| *Diagnosis, n (%)* | |
| Gastric cancer | 31 (91.2) |
| GEJ cancer | 3 (8.8) |
| *Bormann subtype, n (%)* | |
| I | 3 (8.8) |
| II | 18 (52.9) |
| III | 7 (20.6) |
| IV | 6 (17.7) |
| *Clinical T category[a], n (%)* | |
| T3 | 10 (29.4) |
| T4a | 19 (55.9) |
| T4b | 5 (14.7) |
| *Clinical N category[a], n (%)* | |
| N1 | 1 (2.9) |
| N2 | 21 (61.8) |
| N3 | 12 (35.3) |
| *Histologic grade, n (%)* | |
| G2 | 12 (35.3) |
| G3 | 22 (64.7) |
| Signet ring cell carcinoma, n (%) | 4 (11.8) |
| *Lauren's classification, n (%)* | |
| Intestinal type | 17 (50.0) |
| Diffuse type | 10 (29.4) |
| Mixed type | 6 (17.7) |
| Unknown | 1 (2.9) |
| *PD-L1 CPS, n (%)* | |
| <1 | 16 (47.1) |
| ≥1 and <5 | 5 (14.7) |
| ≥5 | 11 (32.4) |
| Unknown | 2 (5.9) |
| *MSI/MMR status, n (%)* | |
| MSI-H/dMMR | 1 (2.9) |
| MSS/pMMR | 33 (97.1) |
| *HER2 status, n (%)* | |
| 0 | 19 (55.9) |
| 1+ | 10 (29.4) |
| 2+ | 5 (14.7) |

*ECOG PS* Eastern Cooperative Oncology Group Performance Status, *GEJ* gastroesophageal junction, *MSS* microsatellite stable, *pMMR* proficient mismatch repair, *CPS* combined proportional score, *PD-L1* programmed cell death ligand 1.
[a]Pathological tumor and lymph-node statuses were classified according to the criteria proposed by the eighth edition of the Cancer Staging Manual of the American Joint Committee on Cancer.

Immune checkpoint inhibitors that target programmed cell death protein 1 (PD-1) or programmed cell death ligand 1 (PD-L1) have shown promising survival benefits and manageable safety in first-line treatment of patients with G/GEJ cancers in the CheckMate-649[10], GEMSTONE-101[11], and ORIENT-16 trials[12]. By binding to PD-1, activating antigen-specific T cells, and reversing the immune evasion of cancer, immunotherapy may be more effective in the early stages of G/GEJ cancers when the tumor is still present after the approach known as neoadjuvant therapy[13]. Although the survival benefit of neoadjuvant immunotherapy has been demonstrated in several solid tumors, including breast cancer[14] and hepatocellular carcinoma[15], lung cancer[16], and melanoma[17], there is limited evidence in G/GEJ cancers.

In recent studies, it has been demonstrated that conventional cancer therapies, including chemotherapy and radiotherapy, alter the tumor immune microenvironment (TiME) (e.g., increasing the expression levels of immune checkpoints[18]), which is crucial for the development, progression, and therapeutic responses of tumors[19]. Besides, preclinical studies indicated the synergistic effect of concurrent chemoradiotherapy (cCRT) on immunotherapy by enhancing the host immune response and inhibiting cancer cell immune escape[20,21]. The CheckMate-577 study demonstrated the survival benefit of adjuvant nivolumab in patients with esophageal/GEJ cancer who received preoperative chemoradiotherapy[22]. However, the efficacy and safety of perioperative immunotherapy plus cCRT in locally advanced gastric cancer remain mostly unexplored[23].

Sintilimab, a fully human and highly selective anti-programmed death (PD)-1 monoclonal antibody, binds to the PD-1 receptor and blocks its interaction with PD-L1 and PD-L2, causing a robust anti-tumor effect[24]. In the ORIENT-16 study, first-line sintilimab plus chemotherapy improved survival in Chinese patients with advanced G/GEJ cancers[12]. There is evidence of the efficacy of paclitaxel in gastric cancer[25] and of nab-paclitaxel in the second- and third-line treatment of gastric cancer[26–28]. There is also evidence of the non-inferiority efficacy of nab-paclitaxel vs. paclitaxel in previously treated advanced gastric cancer[29] and significantly longer progression-free survival of additional nab-paclitaxel vs. paclitaxel to S-1 in the first line treatment of advanced gastric cancer[30]. In addition, nab-paclitaxel does not require premedication with corticosteroids. In Japan, D2 gastrectomy and adjuvant S-1 are the standard management of locally advanced gastric cancer[31]. The combination of nab-paclitaxel and S-1 has been demonstrated as an effective and safe first-line treatment[28].

Hence, in this multicenter, single-arm phase 2 trial, we aim to evaluate the efficacy and safety of perioperative sintilimab plus cCRT in patients with locally advanced G/GEJ cancers. Further, we conduct an exploratory multiplex immunofluorescence (mIF) analysis of TiME with the aim of identifying promising biomarkers of treatment response. Here, we show the promising efficacy and manageable safety profile of sintilimab in combination with concurrent chemoradiotherapy for the perioperative treatment of locally advanced G/GEJ adenocarcinoma.

## Results
### Patient characteristics
From July 20, 2019 to October 10, 2021, 42 patients were screened for eligibility, and 8 were excluded due to refusal to participate (n = 4), peritoneal metastasis (n = 3), and a history of severe rash (n = 1, Supplementary Fig. 1). Finally, a total of 34 patients were enrolled, with a median age of 65.5 years (range, 58–68). Among them, 28 patients (82.4%) were male, and 26 (76.5%) had an ECOG PS score of 0. Thirty-one patients (91.2%) were diagnosed with gastric cancer, and 3 (8.8%) had GEJ cancer. Three (8.8%), 18 (52.9%), 7 (20.6%), and 6 (17.7%) cases were Bormann type I, II, III, and IV, respectively. Detailed patient baseline characteristics are presented in Table 1.

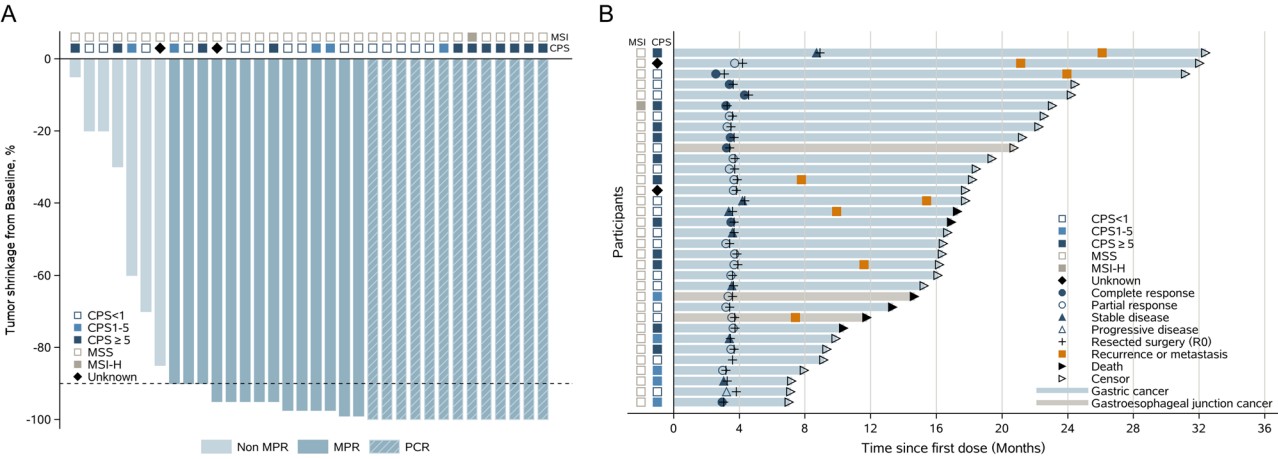

**Fig. 1 | Treatment response (*n* = 34).** Tumor shrinkage from baseline (**A**) and duration of disease response (**B**). All patients had R0 resection. The duration of response was censored at the time of the procedure.

## Pathological responses

All patients received neoadjuvant therapy and underwent surgery. Thirty-three (97.1%) patients underwent total gastrectomy plus D2 lymphadenectomy, and 1 (2.9%) underwent proximal gastrectomy with D2 lymphadenectomy. The rate of R0 resection was 100%. Thirty-one (91.2%) patients underwent adjuvant therapy, while 3 did not due to refusal (*n* = 2) and hepatic toxicity (*n* = 1).

Tumor shrinkage from baseline is shown in Fig. 1A. Thirteen patients achieved pathological complete response (pCR), and the pCR rate was significantly higher than that of the null hypothesis (38.2% vs. 15%, *P* = 0.001). Twenty-seven patients (79.4%) had MPR. Pathological T stages after surgery were ypT0, ypT1b, ypT2, ypT3, and ypT4a in 13 (38.2%), 2 (5.9%), 9 (26.5%), 8 (23.5%), and 2 (5.9%) patients, respectively. Based on pathological N stage after surgery, 24 (70.6%), 2 (5.9%), 3 (8.8%), and 5 (14.7%) patients were ypN0, ypN1, ypN2, and ypN3, respectively (Table 2). The subgroup analysis of pCR is presented in Supplementary Fig. 2. Among the six participants with Bormann IV tumors, the MPR was 83.3% (5/6), the pCR rate was 16.7% (1/6), the rate of R0 resection was 100% (6/6), one was ypT0, one was ypT2, four were ypT3, three were ypN0, two were ypN2, and one was ypN3a.

## Table 2 | Pathologic responses after treatment

| Tumor response | Total (*n* = 34) | |
|---|---|---|
| | *n* (%) | Confidence interval[a] (95%) |
| R0 resection | 34 (100.0) | 0.90–1.00 |
| pCR | 13 (38.2) | 0.24–0.54[b] |
| MPR | 27 (79.4) | 0.62–0.91 |
| *Pathological T stage post-surgery* | | |
| ypT0 | 13 (38.2) | 0.22–0.56 |
| ypT1b | 2 (5.9) | 0.01–0.20 |
| ypT2 | 9 (26.5) | 0.13–0.44 |
| ypT3 | 8 (23.5) | 0.11–0.41 |
| ypT4a | 2 (5.9) | 0.01–0.20 |
| *Pathological N stage post-surgery* | | |
| ypN0 | 24 (70.6) | 0.53–0.85 |
| ypN1 | 2 (5.9) | 0.10–0.20 |
| ypN2 | 3 (8.8) | 0.02–0.24 |
| ypN3 | 5 (14.7) | 0.05–0.31 |

*pCR* pathological complete response, *MPR* major pathological response.
[a]The confidence intervals were estimated by the Clopper-Pearson exact method.
[b]The alpha level for the confidence interval of pCR was 90%.

## Survival outcomes

At the cutoff date of April 8, 2022, median follow-up times were 13.9 (range 1.0–20.9) months for disease-free survival (DFS) and 18.2 (range, 7.0–32.4) months for event-free survival (EFS) and OS. The survival of patients with different treatment responses is summarized in Fig. 1B. The median DFS was 17.0 (95%CI: 11.1–20.9) months (Fig. 2A), with a 1-year DFS rate of 64.5% (95%CI: 30.4–85.1). Patients achieving pCR had significantly longer median DFS than those who did not achieve pCR (20.9 vs. 11.1 months, log-rank *P* = 0.0285, Supplementary Fig. 3A). The median EFS was 21.1 (95%CI: 14.7–26.1) months, and the 1-year EFS rate was 80.1% (95%CI: 40.4–94.7, Fig. 2B). Six patients died during the follow-up period. The median OS was not reached. The 6-month and 1-year OS rates were 100.0% (95%CI: 89.7–100.0%) and 92.6% (95%CI: 50.1–99.5%), respectively (Fig. 2C).

## Safety profile

During the neoadjuvant period, 32 (94.1%), 31 (91.2%), and 11 (32.4%) patients experienced treatment-emergent AEs (TEAEs), treatment-related AEs (TRAEs), immune-related AEs (irAEs), respectively (Table 3). The most common TEAEs during the neoadjuvant period were myelosuppression (*n* = 27, 79.4%), nausea/vomiting (*n* = 17, 50.0%), and rash (*n* = 9, 26.5%). Seventeen patients (50.0%) had grade ≥3 AEs. The most common grade ≥3 AE was myelosuppression (*n* = 11, 32.4%). cCRT-related adverse events occurred in 28 (82.4%) patients, including 10 (29.4%) with grade 3 cCRT-related AEs (7 myelosuppression, 4 nausea/vomiting and 1 with both.) and 4 (11.8%) with grade 4 myelosuppression. The most common cCRT-related AEs were myelosuppression and nausea/vomiting (Supplementary Table 1). Two (5.9%) had radiotherapy discontinuation: one patient due to grade 3 nausea/vomiting at the dose of 36Gy/20f and one due to grade 3 nausea/vomiting and myelosuppression at the dose of 34.2 Gy/19f. The immunotherapy-related AEs (irAEs) are presented in Supplementary Table 2.

No patient postponed surgery due to AEs (Table 3). Thirteen patients (38.2%) experienced surgical AEs, and 1 (2.9%) had grade 3 surgical AEs. The most common surgical AEs were hypokalemia (*n* = 6, 17.7%) and elevated alanine aminotransferase/aspartate aminotransferase (*n* = 5, 14.7%). One (2.9%) patient developed a grade 2 operative complication of anastomotic leakage (Supplementary Table 3). During the adjuvant period, 23 patients (74.2%) had TEAEs, and most of them showed myelosuppression (*n* = 22, 71.0%). Eleven patients (35.5%) had grade ≥3 AEs, all of which were myelosuppression (Table 3).

## Biomarker analysis

Baseline tumor biopsies were available in 94.1% (32/34) of patients. Patients with positive PD-L1 expression showed a numerically higher

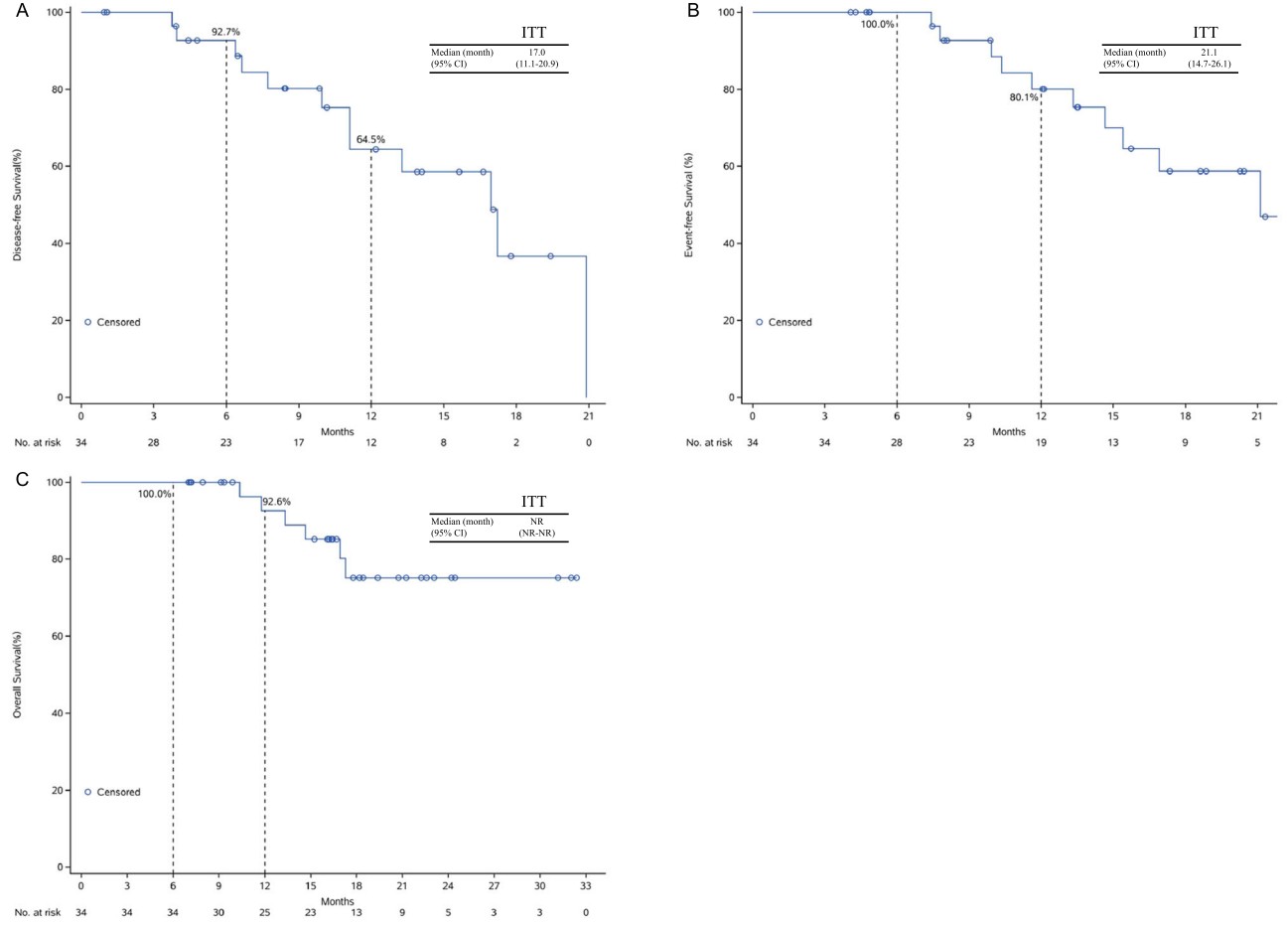

**Fig. 2 | Survival outcomes of all patients.** (**A**) Disease-free survival, (**B**) event-free survival and (**C**) overall survival. Source data are provided as a Source Data file.

pCR rate (combined proportional score (CPS) ≥ 5 vs. CPS < 5, 63.6% vs. 28.6%, $P = 0.072$, Supplementary Fig. 2), while no association was found between PD-L1 expression and DFS (Supplementary Fig. 3B, C). One case was microsatellite-instability (MSI)-H and achieved pCR (Supplementary Fig. 2).

To further assess whether different immune cell subtypes within the TiME could predict treatment response, both baseline tumor biopsies and post-treatment surgical tissues were analyzed by the mIF assay. A total of 47 samples passed quality control and were tested in the final data analysis (23 who underwent gastroscopy and 24 who underwent surgery, including 17 samples that were paired).

Patients achieving pCR had higher levels of infiltrating CD3+ T cells, CD4+ T cells, and CD56+ natural killers (NKs) (CD56bright and CD56dim subtypes) than those not achieving pCR (Fig. 3A–E). There were no significant differences in the levels of infiltrating CD20+ and CD8+ T cells and PD-1 and FoxP3 expression between the two groups (Fig. 3F–I). A significantly higher M1 to M1 + M2 macrophage infiltration ratio was observed in patients achieving pCR (Fig. 3J–L). After adjusting by Bonferroni correction test, the levels of CD56+ and CD56dim cells in stroma were positively correlated with efficacy with an adjusted P-value of 0.024 and 0.012, respectively, while adjusted P-values of other biomarkers were all >0.05 (Supplementary Table 4). Typical mIF images showed higher immune infiltration rates of CD56bright, CD56dim, and CD3 in two patients achieving pCR (Fig. 4A) compared with those not achieving pCR (Fig. 4B).

After neoadjuvant treatment, enhanced CD20+ B cell infiltration was found in patients achieving pCR than in patients not achieving pCR. Besides, a trend that infiltration of CD4+ T cells was higher in patients achieving pCR than that in patients not achieving pCR

(median density: 10.1 vs. 0.4 counts/mm², $P = 0.059$, adjusted $P = 0.708$) was observed. At the same time, there was no obvious significant difference in other biomarkers between the two groups (Supplementary Fig. 4). It should be noted that at baseline, two patients had relatively high levels of CD20+ B and CD3+ T cell infiltration, but TLSs were not detected, while TLSs were detected along with pCR after treatment (Fig. 4C, D). The changes in biomarkers are shown in Supplementary Fig. 5.

## Discussion

The role of immunotherapy plus cCRT in the perioperative treatment of G/GEJ cancers is unknown. In this study, we evaluated the efficacy and safety of neoadjuvant sintilimab and cCRT followed by gastrectomy and adjuvant sintilimab and chemotherapy in patients with locally advanced G/GEJ cancers. Our results met the pre-specified primary endpoint, with a pCR rate of 38.2%. A total of 79.4% of patients achieved MPR, and all patients received R0 resection. The median DFS and EFS were 17.0 months and 21.1 months, respectively. In addition, biomarker analysis showed that the level of CD56+ and CD56dim NK cells in stroma at baseline were associated with pathological response.

In the neoadjuvant setting, the CROSS trial reported a pCR rate of 23% specifically in patients diagnosed with esophagus or GEJ adenocarcinoma, while the POET trial reported a pCR rate of 15.6% in patients with esophagus or GEJ cancer[8,32]. The interim results of the TOPGEAR trial suggest that preoperative radiation and intensive chemotherapy (epirubicin, cisplatin and fluorouracil) is safe for the vast majority of patients without additional treatment toxicity or surgical morbidity[33]. In addition to cCRT, several studies have investigated the role of preoperative immunotherapy-containing therapies in G/GEJ cancer

**Table 3 | Safety profiles**

| Adverse events, n (%) | Any periods (n = 34) | Neoadjuvant period (n = 34) | Surgery (n = 34) | Adjuvant period (n = 31) |
|---|---|---|---|---|
| TEAE | 34 (100.0) | 32 (94.1) | 13 (38.2) | 23 (74.2) |
| TRAE | 34 (100.0) | 31 (91.2) | 13 (38.2) | 22 (71.0) |
| irAE | 11 (32.4) | 11 (32.4) | 0 | 1 (3.2) |
| Grade ≥3 AEs | 22 (64.7) | 17 (50.0) | 1 (2.9) | 11 (35.5) |
| *TEAEs* | | | | |
| Myelosuppression | 33 (97.1) | 27 (79.4) | 0 | 22 (71.0) |
| Nausea/vomiting | 18 (52.9) | 17 (50.0) | 0 | 1 (3.2) |
| Elevated ALT/AST | 11 (32.4) | 5 (14.7) | 5 (14.7) | 2 (6.5) |
| Decreased albumin | 9 (26.5) | 5 (14.7) | 3 (8.8) | 1 (3.2) |
| Rash | 9 (26.5) | 9 (26.5) | 0 | 0 |
| Hypokalemia | 8 (23.5) | 1 (2.9) | 6 (17.7) | 3 (9.7) |
| Fever | 3 (8.8) | 3 (8.8) | 0 | 0 |
| Fatigue | 2 (5.9) | 2 (5.9) | 0 | 0 |
| Pneumonia/ pneumonitis | 2 (5.9) | 1 (2.9) | 1 (2.9) | 0 |
| Upper respiratory tract infection | 1 (2.9) | 1 (2.9) | 0 | 0 |
| Hypopituitarism | 1 (2.9) | 1 (2.9) | 0 | 1 (3.2) |
| Elevated bilirubin | 1 (2.9) | 1 (2.9) | 0 | 0 |
| Elevated creatinine | 1 (2.9) | 1 (2.9) | 0 | 0 |
| Hyperthyroidism | 1 (2.9) | 1 (2.9) | 0 | 0 |
| Urinary tract infection | 1 (2.9) | 1 (2.9) | 0 | 0 |
| Upper gastro-intestinal bleeding | 1 (2.9) | 1 (2.9) | 0 | 0 |
| Anastomotic leakage | 1 (2.9) | 0 | 1 (2.9) | 0 |
| Intestinal obstruction | 1 (2.9) | 0 | 1 (2.9) | 0 |

AEs not indicated in this table had an occurrence of 0.

*AE* adverse event, *TEAE* treatment emergent adverse event, *TRAE* treatment related adverse event, *irAE* immune-related adverse event, *ALT* alanine aminotransferase, *AST* aspartate aminotransferase.

patients. Neoadjuvant nivolumab monotherapy in resectable gastric cancers showed a pCR of only 3.2% and an MPR of 16.7%[34]. A phase II trial investigated preoperative pembrolizumab plus cCRT in 31 patients with GEJ cancer, of whom 7 (22.6%) achieved pCR, which did not meet the pre-specified primary endpoint[35]. Despite these two trials showing suboptimal efficacy, they demonstrated the potential effects of immunotherapy in the neoadjuvant setting for G/GEJ cancers. Likewise, another two single-arm phase II trials[36,37] evaluated the effects of neoadjuvant PD-1 blockade plus chemotherapy or chemoradiotherapy in patients with locally advanced G/GEJ cancers, which showed promising efficacy, with a pCR rate of 28.0% and 33.3%, respectively. The ongoing phase II GASPAR trial of the perioperative FLOT regimen combined with spartalizumab will provide additional data in the future[38]. In addition, the benefit and risk of intensive chemotherapy regimen would be properly discussed when the full results of TOPGEAR study and other ongoing clinical trials that use the combination of FLOT and immunotherapy (e.g. MATTERHORN study, NCT04592913; KEYNOTE-585 study, NCT03221426).

In this study, 13 patients had pCR, whose rate was significantly higher than the null hypothesis (38.2% vs. 15.0%, P = 0.001), as well as higher than those of previous studies[34–37]. Better outcomes may be conferred by the combination of cCRT and immunotherapy. According to preclinical studies, cCRT enhances the immune response of the

host and inhibits the immune escape of cancer cells during immunotherapy[20,21]. This combination was shown to improve the survival of cancer patients in the CheckMate-577 studies, in which cCRT was followed by subsequent immunotherapy[22]. Synergism with PD-1/PD-L1 blockade and radiotherapy was also reported in lung cancer with advanced disease stages[39,40] and early-stage diseases[41]. Besides, the chemotherapy regimen in this study was nab-PTX, which showed promising antitumor activity in the first-line treatment of advanced gastric cancer[30]. As a potential immunomodulator, nab-PTX reverses the immunosuppressive microenvironment and promotes the cancer-immunity cycle in gastric cancer[42], which was demonstrated to be effective in combination with concurrent radiotherapy or immunotherapy in various solid tumors[43,44]. Nevertheless, despite these potential reasons as well as the designed induction phase of immunotherapy plus chemotherapy in this study, the relatively high pCR rate among patients with advanced clinical tumor T stage (T3, 29.4%; T4, 71.7%) and a high proportion of lymph node metastasis (N2-3, 97.1%) demonstrated the encouraging efficacy of sintilimab in combination with cCRT for locally advanced G/GEJ adenocarcinomas, which could be further verified in future studies.

PD-L1 CPS was proven to predict outcomes in metastatic gastroesophageal cancer[10]; however, its predictive value was not determined in early-stage diseases. We found that patients with higher PD-L1 CPS (≥5 vs. CPS < 5) achieved a numerically higher pCR rate (63.6% vs. 28.6%, P = 0.072), consistent with a previous report[35]. However, PD-L1 CPS was not associated with survival outcomes (DFS & OS) in this study. This may be due to a relatively short follow-up time and immature survival data. Patients with MSI-H were reported to benefit from immunotherapy in colorectal and gastric cancers. In this study, one patient was MSI-H and achieved pCR, in line with previous studies[45,46], indicating the potential predictive value of MSI-H, although limited by the sample size.

NK cells play a crucial role in early antitumor immunity by directly killing tumor cells[47]. CD56$^{dim}$ NK cells are the final mature stage of NK cells and have a higher killing ability compared with CD56$^{bright}$ NK cells. Studies have shown higher levels of NK cell infiltration are associated with better tumor outcomes[48]. A better therapeutic response was associated with both CD56$^{dim}$ and CD56$^{bright}$ NK cells in this study. As a result, CD56$^+$ NK cells might be a good predictor of the response to immunotherapy plus cCRT in locally advanced G/GEJ cancers. Tumor-associated macrophages play important roles in tumor immune response, tumor cell proliferation, and tumor invasion[49]. Immune responses can be induced by the transformation of type M2 (pro-tumor type) macrophages into type M1 (antitumor type) macrophages[49]. In this study, patients achieving pCR showed significantly higher M1 to M1 + M2 macrophage infiltration ratio, which was consistent with a previous study[50]. CD20 is a receptor on B lymphocytes, which play an important role in the regulation of the immune system and antitumor activity[51]. The infiltration of CD20$^+$ B cells was enhanced in patients achieving pCR in this study, in agreement with a previous study[52]. Additionally, CD20$^+$ B cells are a strong prognostic factor, and patients with abundant B cells have longer survival after treatment with immune checkpoint inhibitors[53], which was verified by long-term follow-up in this study.

A TLS is an ectopic lymphoid organ induced by chronic inflammation and tumors, mainly composed of CD20$^+$ B cells and CD3$^+$ T cells, which has been associated with better outcomes following treatment with immune checkpoint inhibitors[54]. Studies have shown that chemotherapy and immunotherapy induce TLS formation and B cell aggregation[55,56]. In this study, two patients had TLSs after immunotherapy plus cCRT and achieved pCR, which is consistent with a previous study revealing elevated response rate to melanoma neoadjuvant immunotherapy is associated with TLS detection in on-treatment samples[52]. Among the patient with TLS at baseline, the pCR rate was 40% (2/5), similar to 38.2% in the whole cohort. The MPR rate was 80%, also

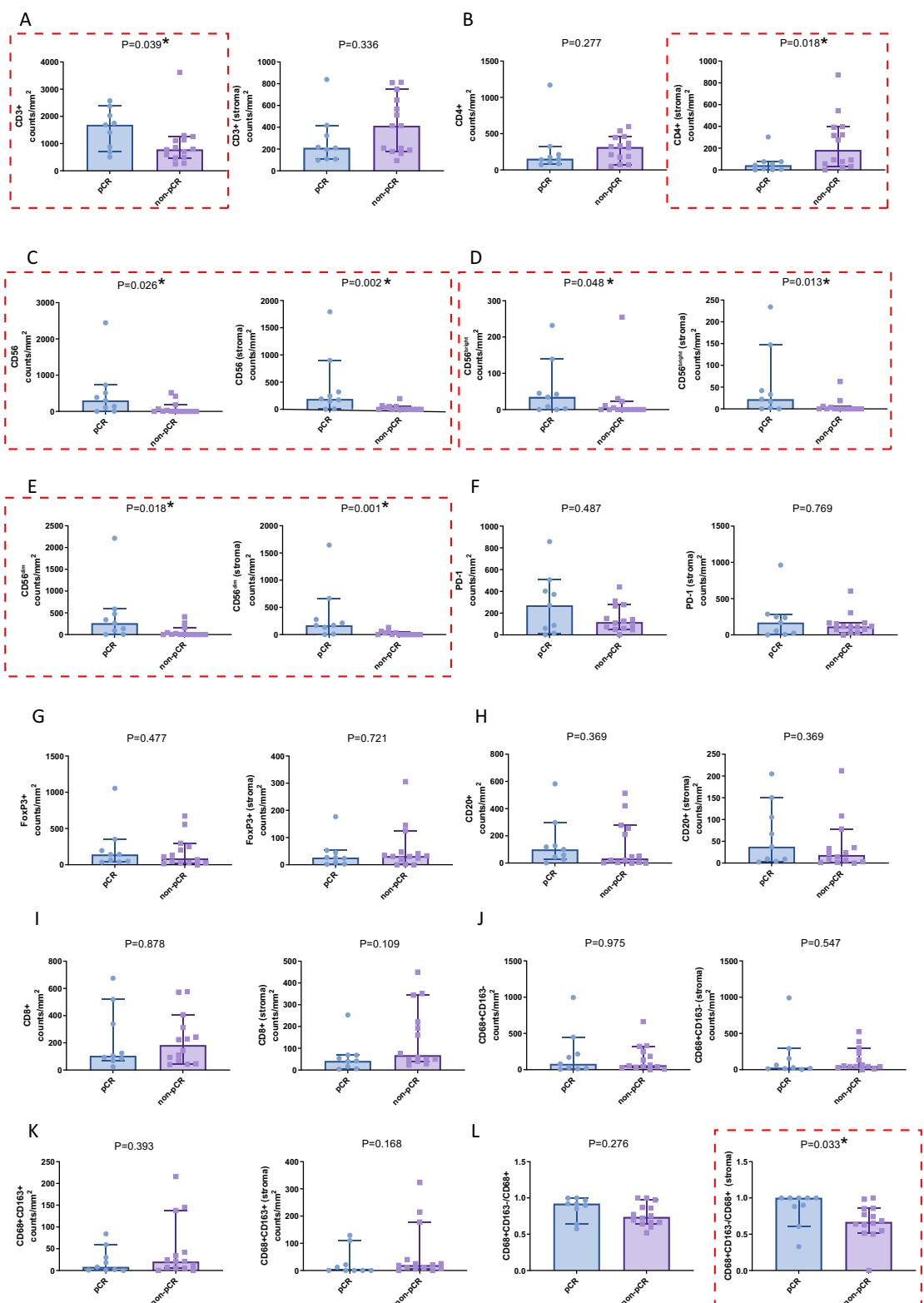

**Fig. 3 | Immune cell infiltration levels in the tumor tissue before neoadjuvant sintilimab and concurrent chemoradiotherapy between patients achieving pCR (_n_ = 9) and those not achieving pCR (non-pCR; _n_ = 14) assessed by multiplex immunofluorescence (mIF). A–L** The comparison of immune cell infiltration levels was performed using Friedman's non-parametric test and the adjusted _P_-value using Bonferroni methods were presented in Supplementary Table 4. The error bars represented the standard deviation. Indicators marked with red dashed boxes represent _p_ < 0.05. Source data are provided as a Source Data file.

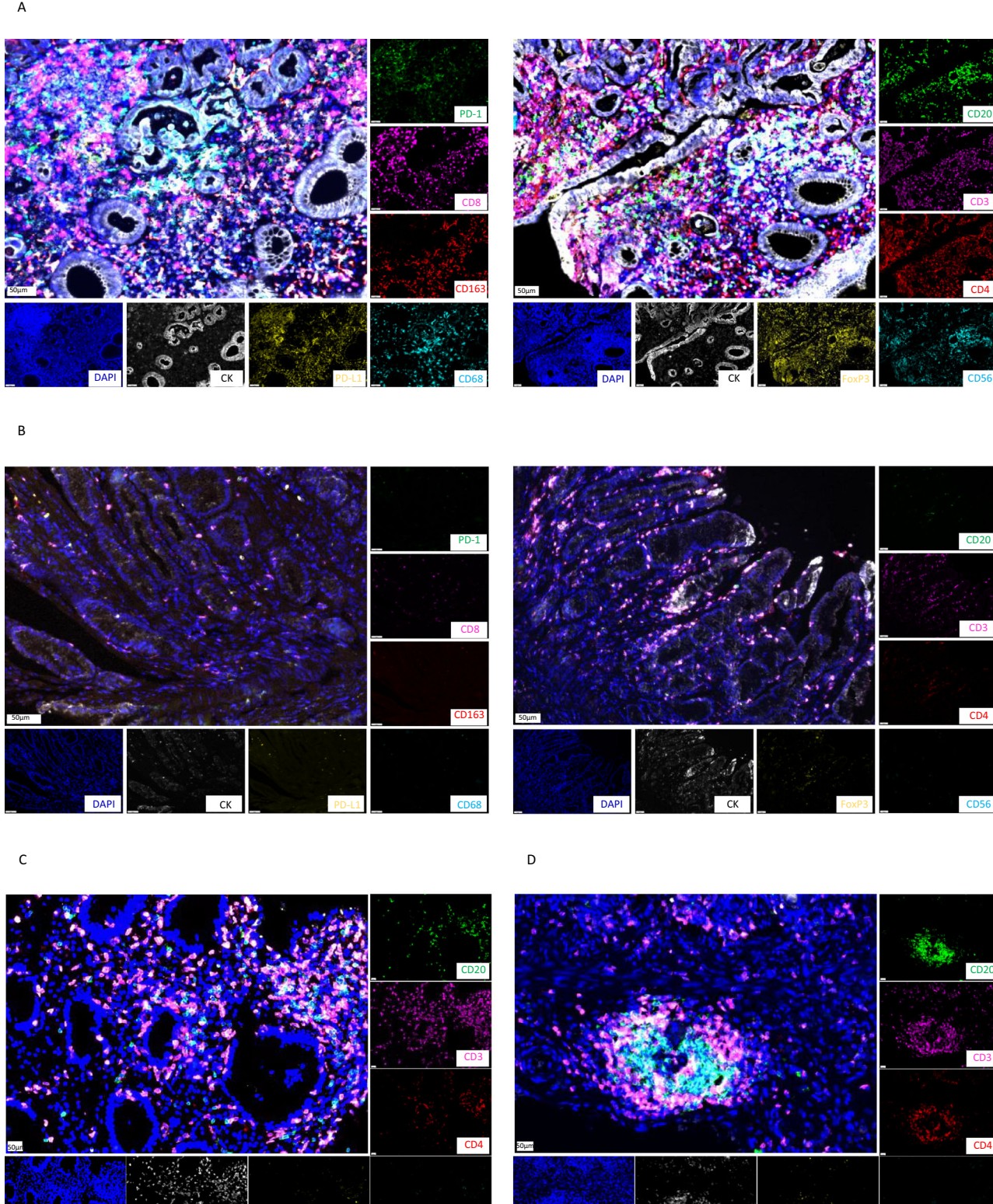

**Fig. 4 | Representative multiplex immunofluorescence (mIF) of tumor immune microenvironment (TiME). A**, **B** Typical mIF images showing elevated CD56$^{bright}$, CD56$^{dim}$ and CD3 cell infiltration in a patient achieving pCR (**A**) than another patient not achieving pCR (**B**). **C**, **D** Typical mIF images in a patient achieving pCR with relatively high levels of CD20$^{+}$ B and CD3$^{+}$ T cell infiltration at baseline (**C**) and developed tertiary lymphoid structure after therapy (**D**). The images are representative of three patients, bar = 50 um.

similar to 79.2% in the whole cohort. It is possible that endoscopy biopsy does not cover large tumor stromal areas rich in TLS structure, and the number of TLSs will be underestimated[57]. According to these findings, immunotherapy combined with cCRT for locally advanced G/GEJ cancers modulates the TiME, including triggering TLS generation, which might be associated with improved survival and response.

Regarding survival, the 6-month and 1-year OS rates in this study were 100.0% and 92.6%, respectively. One-year OS rates of 79.8% and 75% were reported for patients with GEJ cancers using neoadjuvant cCRT in the CROSS and POET studies, respectively[8,32]. Besides, we found that patients with pCR had longer median DFS than those with Non-pCR (20.9 vs. 11.1 months), consistent with a previous report[32]. Considering the relatively short median follow-up time (18.2 months, range: 7.0–32.4) and immature OS data, whether the improved pCR could translate to survival benefit in this trial should be further investigated.

The most common TEAE during the neoadjuvant period in this study was myelosuppression, followed by nausea/vomiting and rash, corroborating previous studies[36]. A total of 17 patients experienced manageable grade ≥3 AEs, of which the most common was myelosuppression. One patient developed a grade II operative complication of anastomotic leakage due to improper eating, which was also reported in a previous study in which anastomotic leakage occurred in 10.3% (3/29) of patients[35]. The case in this study recovered seven days later upon fasting and symptomatic treatment and did not undergo a second operation. There was no additional safety concern, suggesting the feasibility of sintilimab in combination with cCRT.

Nab-paclitaxel was selected in the present study for several reasons. First, there is evidence of the efficacy of paclitaxel in gastric cancer[25,58–60] and nab-paclitaxel in the second- and third-line treatment of gastric cancer[26–28]. Second, there is also evidence of the non-inferior efficacy of nab-paclitaxel vs. paclitaxel in advanced gastric cancer[29]. In Japan, nab-paclitaxel is approved for the second-line treatment of gastric cancer[26,27]. Nab-paclitaxel has been shown effective as a first-line treatment[28] and demonstrated longer progression-free survival than paclitaxel in first-line treatment for advanced gastric cancer[30]. Third, nab-paclitaxel has a synergistic effect with immunotherapy[61,62]. In addition, nab-paclitaxel has a synergistic effect with radiotherapy in lung cancer, pancreatic cancer, and head and neck squamous cancer, among others[43,63–65]. Fourth, nab-paclitaxel does not require pre-medication with corticosteroids, which might be more suitable in the context of the unsure impact of corticosteroids in patients receiving immunotherapy. Although there is no compelling evidence that corticosteroids decrease the efficacy of immunotherapy[66], corticosteroids and immunotherapy both influence the immune system in different ways[67], and it was considered prudent when designing the trial to avoid that potential confounder.

In the present study, radiotherapy was given according to local physicians' preferences, according to guidelines, experience, and tumor board discussions. Future studies could look into specific radiotherapy regimens for preoperative sintilimab in combination with cCRT. Different doses and fractionation regimens could yield different results and should be explored in the future.

This study had some limitations. This was a single-arm trial with a limited sample size, and no control group was included. Although this study showed promising short-term outcomes (e.g., pCR rate) and middle-term outcomes (e.g., DFS and EFS), the long-term efficacy of the combination perioperative therapy was not assessed. A follow-up study is ongoing, and survival data will be reported in the future. This study was a preliminary exploratory phase II trial, and selecting the pCR as the primary endpoint is common in such trials of solid tumors as it provides an answer more rapidly than survival[68–70]. In the RTOG 9904 trial, the patients with pCR achieved better outcomes than the non-pCR patients, and pCR was suggested as a prognostic surrogate[69,71]. Future studies will have a longer follow-up. Only one case of grade 2 fatigue

was reported in our study, a finding which seems to diverge from expectations. The relatively small sample size may have led to an incomplete capture of patient experiences. Additionally, the potential for less stringent reporting and observation practices of adverse reactions such as fatigue could have been a contributing factor. Fatigue is inherently subjective, and thus its accurate assessment can be challenging. Finally, this study explored biomarkers, but the sample size does not allow for conclusions. The biomarker results could be referred to for the design of future large-scale trials.

In this multicenter, single-arm trial, sintilimab in combination with cCRT demonstrated promising efficacy and a favorable safety profile in the perioperative setting for locally advanced G/GEJ adenocarcinomas. Further large-scale randomized clinical trials are warranted to confirm the survival benefit.

## Methods

### Study design and participants

The SHARED study (ChiCTR1900024428) followed the Declaration of Helsinki and Good Clinical Practice guidelines and was approved by the ethics committees of the Comprehensive Cancer Centre of Drum Tower Hospital (2019-093-02) and other participating centers. All patients provided written informed consent before any procedure.

This study was a multicenter, single-arm phase 2 trial conducted at three centers in China (The date of registration was July 11, 2019, https://www.chictr.org.cn/showprojEN.html?proj=40332). The study protocol was published previously[72]. Treatment-naïve patients aged over 18, with histologically or cytologically confirmed locally advanced G/GEJ cancers (cT3N2-3M0, cT4aN + M0, or cT4bNanyM0, determined by computed tomography and/or magnetic resonance imaging before treatments), an Eastern Cooperative Oncology Group performance status (ECOG PS) score of 0-1, and at least one measurable lesion based on Response Evaluation Criteria in Solid Tumors (RECIST) v1.1 were enrolled from July 20th 2019 to October 10th 2021.

Patients with gastric neuroendocrine tumors, distant metastasis, uncontrolled pleural effusion, pericardial effusion, ascites, history of chemotherapy, radiotherapy, or immunotherapy, history of cancer within the past 5 years (except basal cell or squamous cell skin cancer, superficial bladder cancer, in-situ cervical cancer, or in-situ breast cancer), gastrointestinal obstruction, uncontrolled infection, uncontrolled systemic disease, or the use of immunosuppressive agents or experimental drugs in the past 4 weeks were excluded. Patients with severe cardiovascular diseases, such as symptomatic coronary heart disease, grade II congestive heart failure, uncontrolled arrhythmia and myocardial infarction, occurred within 12 months before admission; or with a history of interstitial pulmonary disease, non-infectious pneumonitis, pulmonary fibrosis, and acute pulmonary disease; or respiratory condition that required any oxygen supplementation, active pneumonitis and clinically significant pulmonary hypertension occurred within 12 months before admission were excluded. Pregnant, breastfeeding, or pregnancy test-positive women were also excluded[72].

### Procedure

Eligible patients received one 3-week cycle of induction treatment consisting of S-1 (40 mg/m$^2$, PO, bid, D1-14), nab-paclitaxel (nab-PTX, 100–120 mg/m$^2$, IV, D1, and 8), and sintilimab (200 mg, IV, D1), followed by 5 weeks of radiation therapy (45 Gy/1.8 Gy in 25 factions), nab-PTX (80–100 mg/m$^2$, IV, D1, 8, 15, and 22) and sintilimab (200 mg, IV, D1, 22). Patients were administered another 3-week cycle of S-1 (40 mg/m$^2$, PO, bid, D1-14), nab-PTX (100–120 mg/m$^2$, IV, D1 and 8) plus sintilimab (200 mg, IV, D1) one to three weeks later, and surgery was preferably scheduled within 1-3 weeks. Four to six weeks after surgery, three 3-week cycles of S-1 (40 mg/m$^2$, PO, bid, D1-14), nab-PTX (100–120 mg/m$^2$, IV, D1 and 8) plus sintilimab (200 mg, IV, D1) were administered as adjuvant therapy[72]. Dose modification of sintilimab was not permitted, and treatment was discontinued in case of disease

progression, death, or unacceptable toxicities. Patients experiencing grade IV leukopenia with fever were administered prophylactic anti-infective therapy with broad-spectrum antibiotics.

Prior to radiotherapy, computed tomography (CT) simulation set as axial scanning with a layer thickness of 3 mm, complemented with abdominal compression, four-dimensional CT, and respiratory gating technologies, was performed to manage respiratory movement and accurately localize the target area. The gross tumor volume (GTV) included primary tumors and metastatic lymph nodes. The clinical target volume (CTV) included GTV and high-risk lymphatic drainage area. The planed target volume (PTV) included CTV with a 5-mm expansion in all direction. The radiotherapy treatment plan included a total dose of 45 Gy, typically administered in 25 fractions over five weeks. Radiotherapy was delivered using conformal, intensity-modulated, or spiral tomographic intensity-modulated techniques, with photon radiation typically at an energy level of 6 MV or 8 MV for conformal radiotherapy and 6 MV for intensity-modulated radiotherapy. The radiotherapy process was meticulously overseen by a multidisciplinary quality assurance team.

A multidisciplinary team assessed the patients' conditions and decided on the surgical protocol according to guidelines[5,73], experience, and tumor board discussions. The technical details of surgery and radiotherapy were presented as Supplementary Note in the Supplementary information file.

Tumor assessments were evaluated per RECIST v1.1 by CT/magnetic resonance imaging (MRI) and endoscopic ultrasound (EUS) every six weeks preoperatively, every nine weeks postoperatively, and every three months after treatment completion until disease progression (up to two years) or new anticancer treatment initiation. Routine blood tests and blood biochemistry were reviewed weekly. Tumor biomarkers and thyroid function were reviewed preoperatively at weeks 3 and 8 of neoadjuvant therapy and postoperatively every 3 weeks. Adverse events (AEs) from the first dosing of the study regimen to 90 days after the last dosing were recorded and graded according to the National Cancer Institute's Common Terminology Criteria for Adverse Events (NCI CTCAE, version 4.03). Surgical complications were grading using Clavien-Dindo. Judgment and grading of the AEs were based on the time of occurrence of the AEs, clinical manifestations, and relevant test and examination results. If necessary, multidisciplinary discussions were held with relevant departments such as imaging, immunology, gastroenterology, and respiratory.

### Endpoints

The primary endpoint was the pCR rate, i.e., the absence of viable residual tumor cells in the resected specimen. Secondary endpoints included major pathological response (MPR, i.e., residual tumor cells below 10% in the resected specimen), R0 resection rate (i.e., complete removal of the tumor with a tumor-free margin), DFS, EFS, and OS. DFS was defined as the time from surgery to postoperative recurrence or death from any cause, whichever occurred first. DFS was censored on the last tumor assessment date for patients still alive and without recurrence. EFS was the time from enrollment to recurrence or death from any cause. EFS was censored on the last tumor assessment date for patients still alive and without recurrence. OS was the time from enrolment to death from any cause. OS was censored on the last date known to be alive for patients without documentation of death. Safety endpoints included TEAEs, TRAEs, irAEs, and surgical AEs (i.e., complications occurring during or within 30 days of surgery).

### Biomarker analysis

PD-L1 expression was assessed in formalin-fixed paraffin-embedded tumor samples before treatment by the PD-L1 immunohistochemistry (IHC) 22C3 pharmDx assay (Dako, Glostrup, Denmark). PD-L1 evaluation was performed using the CPS, determined as the number of PD-L1-positive cells—tumor cells, lymphocytes, and macrophages—divided by the total number of tumor cells × 100. PD-L1 positivity was defined as CPS ≥ 1 or 5. Mismatch repair (MMR) status was assessed locally by polymerase chain reaction (PCR) amplification and fragment analysis or IHC analysis of the DNA mismatch repair proteins MLH1, MSH2, MSH6, and PMS2. Monoclonal antibodies against MLH1 (Clone: ES05, dilution 1:100, Dako Denmark A/S, Denmark), PMS2 (Clone: EP51, dilution 1:100, Dako Denmark A/S, Denmark), MSH2 (Clone: FE11, dilution 1:100, Dako Denmark A/S, Denmark), MSH6 (Clone: EP49, dilution 1:150, Dako Denmark A/S, Denmark) were incubated with tumor sections in a humidified chamber at 4 °C overnight. Negative controls (without the primary antibody) and positive controls were included in each run to ensure the specificity and accuracy of the staining procedure. MSI analysis was performed using the Revised Bethesda Guidelines for two mononucleotides (BAT 25 and BAT 26) and 3 dinucleotides (D2S123, D5S346, and D17S250) microsatellite markers. Tumors were classified as MSI-H/dMMR with 2 or more unstable markers[74] or with no expression of MMR proteins.

The biomarker panel was decided by the trial committee through discussion when designing the trial based on guidelines[5,6] and experience. This panel of 10 biomarkers was selected based on the efficacy of immunotherapy in solid tumors, including biomarkers for T cells (CD3, CD4, and CD8)[75,76], Tregs (FoxP3)[77], B cells (CD20)[52], NK cells (CD56)[47], macrophages (CD68, CD163)[78], and immune escape (PD-1/L1)[46]. The TiME was examined on baseline tumor biopsies and surgical specimens. The Akoya OPAL Polaris Seven-Color Automation multiplex immunohistochemistry panels were applied for mIF staining following the manufacturer's instructions. Primary antibodies used were raised against CD163 (Abcam, ab182422, 1:500), CD8 (Abcam, ab178089, 1:200), CD68 (Abcam, ab213363, 1:1000), CD20 (DAKO, L26, IR604, 1:1), CD3 (DAKO, A0452, 1:1), CD4 (Abcam, ab133616, 1:100), CD56 (Abcam, ab75813, 1:1000), and FoxP3 (Abcam, ab20034, 1:100) for panel 2. Nucleic acids were stained with DAPI. Tumor parenchyma and stroma were differentiated according to pan-CK staining (Abcam, ab7753, 1:100). Digital image analysis was performed with the APTIME image analysis software (3D Medicines). Tertiary lymphoid structures (TLSs) were defined as CD20+ B cell aggregates surrounded by accumulated CD3+ T cells. The density of various immune cell subsets was expressed as the count of positively stained cells per square millimeter (cells/mm²). The total density was calculated by dividing the count of tumor and stroma cells by the area of the tumor and stroma.

### Statistical analysis

A Simon 2-stage design was employed. The null hypothesis was a pCR rate of 15% was determined based on our clinical experience and the pCR rate of 16% reported in the FLOT4-AIO trial[79], as chemotherapy was the standard of care. The alternative hypothesis was a pCR rate of ≥35% in this study. Using a power of 80% and α = 0.05, a minimum of nine patients were needed in stage I, and if a pCR was confirmed in one or two of them, an additional 25 patients were enrolled in stage II. The analysis plan was designed before conducting the trial. Descriptive statistics were primarily used. Continuous data were described as median (range), and categorical data were described as frequency (percentage). The primary endpoint of pCR with a 90% confidence interval (CI) in the whole population and 95% CIs in subgroups were estimated using the Clopper-Pearson method. A binomial test was performed for pCR with the null hypothesis of pCR as 0.15. For MPR, R0 resection, pathological T stages, and pathological N stages, the 95% CIs were estimated by the Clopper-Pearson exact method without statistical test. Time-to-event variables (DFS, EFS, and OS) were analyzed by the Kaplan-Meier method and compared by the Log-rank test. The hazard ratio (HR) and 95% CIs were estimated using the Cox proportional hazards model. In the exploratory analysis of pathological response, associations of categorical variables were analyzed via Chi-square or Fisher Exact test. Continuous data between groups of pathological response were compared by the Mann–Whitney $U$ test.

Bonferroni correction test was used for multiplicity corrections in biomarker analysis. All statistical analyses were performed with SAS version 9.4 (SAS Institute Inc., Cary, NC, USA). Two-sided $P < 0.05$ was considered statistically significant.

### Reporting summary

Further information on research design is available in the Nature Portfolio Reporting Summary linked to this article.

## Data availability

Patient baseline clinical data are available in Table 1 and within the text. The surgery protocol and radiotherapy protocol are available as Supplementary Note in the Supplementary information file. Study Protocol is available in Ref. 72. The individual de-identified participant data, Statistical Analysis plan and the full image dataset are available for scientific purpose by sending requests to the corresponding author Baorui Liu (baoruiliu@nju.edu.cn) within 5 years after this paper's publication. The remaining data are available within the Article, Supplementary Information, or Source Data file. Source data are provided with this paper.

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

## Acknowledgements

The authors thank the participating patients and their families and the study centers and investigators for their contributions to the study. The authors also thank Mengmeng Tang, Xin Zhang, Yue Wang and Yipeng Zhang for their assistance in data collection and sample checking. This trial was supported by Funding for Clinical Trials from the Affiliated Drum Tower Hospital, Medical School of Nanjing University (2022-YXZX-ZL-01) and Beijing Xisike Clinical Oncology Research Foundation (Y-Young2023-019). All investigators received no remuneration. Innovent Biologics (Suzhou) Co., Ltd. provided sintilimab but had no role in the study design or writing of the manuscript.

## Author contributions

B.R.L., W.X.G., and J.W. conceived and designed the study. B.R.L., W.X.G., J.W., and H.T.Y. were responsible for patient recruitment. Q.L., J. Yang, Y.Y., C.Z., and N.Q.D. were responsible for patient treatment and patient care. X.F.L., M.W., L.M.Z., and P.Z. were responsible for surgery. X.S.F., Y.F., L.L., and F.C.L. helped with pathological evaluation. S.L. helped with image evaluation. J. Yan was responsible for quality control of radiotherapy. X.Y.Z. collected the data. Y.Z. and J.W.Z. analyzed and interpreted clinical data. H.C., S.Q.C., and X.C.Z. were responsible for biomarker analysis. JW wrote the first draft of the manuscript.

## Competing interests

H.C., S.C., and X.Z. are employees of 3D Medicines Inc. (Shanghai, China). The remaining authors have no conflicts of interest to declare.

## Additional information

Jia Wei ©[1,2], Xiaofeng Lu[3], Qin Liu[1,2], Yao Fu ©[4], Song Liu[5], Yang Zhao[6], Jiawei Zhou[6], Hui Chen[7], Meng Wang[3], Lin Li[4], Ju Yang[1,2], Fangcen Liu[4], Liming Zheng[3], Haitao Yin[8], Yang Yang[1,2], Chong Zhou[8], Ping Zeng[3], Xiaoyu Zhou[9], Naiqing Ding[1,2], Shiqing Chen[7], Xiaochen Zhao[7], Jing Yan[1,2], Xiangshan Fan[4], Wenxian Guan[3] ✉ & Baorui Liu ©[1,2] ✉

[1]The Comprehensive Cancer Center of Nanjing Drum Tower Hospital, Affiliated Hospital of Medical School, Nanjing University, Nanjing 210008, China. [2]Clinical Cancer Institute of Nanjing University, Nanjing 210008, China. [3]Department of General Surgery, Nanjing Drum Tower Hospital, Affiliated Hospital of Medical School, Nanjing University, Nanjing 210008, China. [4]Department of Pathology, Nanjing Drum Tower Hospital, Affiliated Hospital of Medical School, Nanjing University, Nanjing 210008, China. [5]Department of Radiology, Nanjing Drum Tower Hospital, Affiliated Hospital of Medical School, Nanjing University, Nanjing 210008, China. [6]Department of Biostatistics, Nanjing Medical University, Nanjing 210029, China. [7]Medical Affairs, 3D Medicines Inc, Shanghai 201114, China. [8]Department of Radiotherapy, Xuzhou Central Hospital, Xuzhou 221009, China. [9]Nanjing Drum Tower Hospital Clinical College of Traditional Chinese and Western Medicine, Nanjing University of Chinese Medicine, Nanjing 210023, China. ✉e-mail: guan-wx@163.com; baoruiliu@nju.edu.cn

