## [Peer Review File · Nature Communications]

Neoadjuvant Sintilimab in Combination with Concurrent Chemoradiotherapy for Locally Advanced Gastric or Gastroesophageal Junction Adenocarcinoma: a single-arm phase 2 trialREVIEWER COMMENTS

Reviewer #1 (Remarks to the Author): with expertise in gastric cancer, clinical

This manuscript describes the results of a multi-centre (three centres), single-arm, phase 2 trial in which patients with locally advanced adenocarcinoma of the stomach or GOJ ranging from T3 N2 to T4a and node positive or T4b irrespective of nodal involvement were treated with neoadjuvant chemotherapy plus anti-PD-1 followed by CRT/anti-1 and then adjuvant anti-chemo/anti-PD-1. The primary endpoint of the study was pCR and high rates of pCR were observed compared to historic control of chemotherapy. The high rates of pCR are encouraging, but it's unclear whether these relate to chemotherapy plus anti-PD-1, CRT plus anti-PD-1 or the combination of chemotherapy plus CRT (plus or minus anti-PD-1).

The authors might consider addressing the following comments.

Introduction

- Line 52: the listed 'standard surgical option' is not entirely applicable for all cancers within the described patient group
- Line 62: Checkmate-649 used a PD1 inhibitor so this sentence (i.e. describing efficacy of PD-1 and PD-L1 inhibitors in CheckMate-649) is not correct
- Needs to be more nuanced in the discussion of nab-paclitaxel and its place in the treatment of this malignancy as this is not a component of the standard of care perioperative regimens
- Line 82: Gives the impression that durvalumab is added to neoadjuvant CRT based on CM577 – it is not, it is an adjuvant monotherapy.

Methods:

- Unclear from this document whether any pre-treatment physiological exclusion criteria? E.g renal function assessment? Lung function assessment?
- These are mentioned in protocol so needs to more explicitly point to this as a references. Same goes for the staging investigations and physiological tests which were required pre-treatment?
- The greatest concern, considering that this is a chemoradiotherapy study, is that the

radiotherapy delivery is not described in sufficient detail and the quality control of RT is not clear. For example it is:

- o Unclear what radiotherapy simulation was performed
 - o Unclear where RT delivered to – very scanty details
 - o Presumably delivered 5 fractions per week?
 - o What was the target volume?
 - ♣ Entire stomach? What if GOJ – to what point then?
 - ♣ Regional lymph nodes? Which?
 - ♣ Margins used?
 - o How was target volume grown to CTV and PTV?
 - o On-treatment imaging used? Any immobilisation?
 - o What were the OARs? To what tolerance?
 - o What was the technique?
 - ♣ 3D or 4D?
 - ♣ Conformal or IMRT/VMAT?
 - o What happened if treatment interrupted? Catch up to Rx within normal time?
 - o Was there a standard RT protocol?
 - o Any QA used?
 - o Were patients reviewed whilst on-treatment? At what interval?
 - o How were adverse events defined as relating to radiotherapy rather than other components of the treatment?
- The same can be said for surgery. Please describe in detail which surgical was performed?
- Please also provide the following information:
- How did they choose their panel of biomarkers to evaluate?
 - How did they decide which component of therapy to attribute adverse events to?
 - Was there an IDMC to decide whether to progress from first to second stage of trial design?
 - Any adjustment for multiple significance testing in the biomarker analysis groups?
 - Was there an a priori stats analysis plan?
 - Why did they drop the S-1 during CRT?
 -

Results

- Line 98: 'a total of 34 patients were included in the analysis' – do they mean enrolled?

Makes it sound like some were treated but not analysed

- Line 105: all patients completed neoadjuvant therapy and underwent surgery

o Not actually true

- Patient group:

o No known HER2 status

- Radiotherapy-related adverse events in 65%; 24% with grade 3 and 1 with grade 4.

- Two patients had RT discontinuation

o Supp table 1 indicates one due to a G1 AE and one due to a G3 AE – need extra info on those

o At what point was treatment discontinued? What dose had these patients received? What were the AE? Seems strange that a G1 AE led to discontinuation

- All RT delivered within planned treatment time? Any interruptions or catch up required?

How was catch up done?

- Says all patients completed neoadjuvant therapy and underwent surgery

o BUT – two patients discontinued radiotherapy due to rAEs, so this can't be right

- Toxicity from RT:

o Rates of fatigue seem remarkably low

♣ Only 1 grade 2 fatigue – 67% of CROSS had fatigue of at least grade 1 in severity

o Pneumonia quoted but concern is pneumonitis – was this in the GOJ group?

o No skin tox?

o What is 'gastrointestinal discomfort'?

o Would want specific yes/no to usual list of toxicities expected e.g. as per SCOPE/CROSS GOJ cases

- General toxicity:

o 9% rate of fever in the neoadjuvant period ◊ compared with 80% myelosuppression?

Seems very low but unclear when bloods were taken and how often

o One patient had intestinal obstruction? Did they then not receive adjuvant rx?

o What is gastrointestinal discomfort?

- Unclear long-term follow-up:

o Any post-op cardiac tox? Lung tox?

- o Adverse events only studied up to 90 days post-completion of last bit of IO
- ♣ Treatment ended at 13 weeks post-op \diamond so 3 + 3 months. Should have captured pneumonitis but not long enough for cardiac sequelae so needs commenting on
 - Need to better report surgical complications using Clavien-Dindo grading
 - Two patients discontinued RT:
 - o Why? What was the rrAE that led to them discontinuing?
 - o What dose did they receive by the time they discontinued?
 - Anything about functional hyposplenism?
 - o Did they cover with abx for this?

Translational research,

The investigators have examined a few biomarkers but it's largely descriptive. As pre and post samples are available, it would be interesting to look at the change in biomarker expression on a per patient level.

Discussion

Notes that CROSS found a pCR of 29.2% and POET of 15.6% in dCRT groups

o CROSS actually found a pCR of 23% in patients with adenocarcinoma (i.e. of relevance to this description) so could be better written to help uninformed reader

Would be helpful to correlate more to existing literature with respect to concurrent immunotherapy with CRT, as is an important question in the field. Would also help to explain significance of biomarker analyses. Ideally they should mention TOPGEAR but also need to discuss FLOT/FLOT-IO etc

Further study

o Worth pointing to the potential benefits of different fractionation regimes and how perhaps intensified systemic with shorter course RT might work etc

Reviewer #2 (Remarks to the Author): with expertise in gastric cancer, clinical

Thank you for the opportunity to review this original manuscript by Jia Wei, et al. entitled, "Perioperative sintilimab in combination with concurrent chemoradiotherapy for locally advanced gastric or gastroesophageal junction adenocarcinoma." This is a multicenter, phase 2 trial evaluating the efficacy and safety of preoperative sintilimab plus nab-PTX/S-1 combined with radiotherapy, followed by radical surgery and adjuvant sintilimab plus nab-PTX/S-1. Among 34 patients, all patients underwent gastrectomy with a 100% R0 resection rate and 38.2% pCR rate. TEAEs, TRAEs, irAEs and surgical AEs were reported. The trial was well conducted and achieved pCR rate was impressive, here are the comments:

1. What is the rationale for selecting the pre-operative chemotherapy regimen? The recommended regimens were usually consisting of 5-Fu and platins in China.
2. What is the rationale for setting reference pCR rate as 15%? In Cross trial the pCR rate was 29%, and in the trials conducted by Ajani, the pCR rates were reported 19.5%, 26% and 30% in gastric cancer after nCRT respectively (JCO 2005, JCO 2006, JCO 2004).
3. In trials studying ICIs in perioperative therapy of gastric cancer, the cycles of ICIs were much more than it in this trial which only included three cycles of sintilimab. For example, in Keynote-585, 3 cycles of pembrolizumab with chemo followed by 11 cycles of pembrolizumab alone, in DANTE trial, 4 cycles of atezolizumab with chemo followed by 8 cycles of atezolizumab alone. Please discuss the difference.
4. The aim of this study was to evaluate the efficacy and safety of perioperative therapy, but the primary endpoint was pCR which only indicates the efficacy of preoperative therapy. For perioperative therapy, DFS is more reasonable. I can understand the difficult to conduct such a trial with survival outcome as primary endpoint, some discussion about this topic will be preferred.
5. Could the authors provide more details of irAEs?
6. There were 6 patients with Bormann IV tumor, how about treatment responses in these tumors?
7. One patient received proximal gastrectomy, this type of gastrectomy was controversial in locally advanced gastric cancer and with little evidence in post-neoadjuvant therapy patients. The KLASS05 trial, a randomized trial comparing proximal gastrectomy (PG) with

total gastrectomy (TG), found more advanced pN stage in TG than in PG, which indicated the proximal gastrectomy may have limitation in lymph node dissection. So, proximal gastrectomy should be performed with caution.

8. To my knowledge TLS was more likely to present in the border of submucosa and muscle layer, endoscopy biopsy was usually superficial and may underestimate the abundance of TLS. Among the patient with TLS in baseline, pCR rate was 40% (2/5) similar to 38.2% in whole cohort. MPR rate was 80% similar to 79.2% in whole cohort too. Please revise the discussion more precisely.

9. In Figure 3A, the density of CD3+ T cell was dramatically different between whole tumor and stroma in pCR samples, I am wondering the reason why?

Reviewer #3 (Remarks to the Author): with expertise in biostatistics, clinical trial study design

The authors have completed and presented a one-arm study that met the primary objective with a pCR rate greater than hypothesized.

The data is appropriately analyzed with proper interpretation and conclusions presented.

The methods are missing details that would allow others to replicate the work presented.

The manuscript would be improved with the following revisions:

1) Endpoint definitions of time-to-event endpoints need to include the definition that was used for censoring for each endpoint (DFS, EFS, and OS) separately.

2) Table 2 presents many endpoints that are binary with the count, proportion, and a confidence interval. The primary endpoint is clearly presented with a label for the type of confidence interval and the level. Please add to the methods and the table the method and alpha for all of the other confidence intervals presented.

3) The methods for analysis of the pCR outcome do not include enough detail to produce Supplement Figure 2. Please add the methods for the confidence intervals included.

Additionally, the figure would benefit with presentation with less precision of the estimates.

Considering the sample size, 2 significant digits is sufficient. Additionally, the title of the graph portion of the figure should include that it is proportion with pCR.

4) The analysis of Disease-Free-Survival presented in supplement Figure 3 presents Kaplan-Meier Figures. There is a p-value presented under a hazard ratio on a figure estimated with non-parametric Kaplan-Meier methods. It is unclear if the p-value represents a test using parametric or non-parametric methods. Please clarify what the p-value is testing.

Additionally, consider removing the hazard ratios from the figure due to the mix of methods in each figure.

Reviewer #1

Introduction

- *Line 52: the listed ‘standard surgical option’ is not entirely applicable for all cancers within the described patient group*

Response: We thank the Reviewer for the comment. We revised the statement in the Introduction: “According to the guidelines of the Chinese Society of Clinical Oncology (CSCO), a combination of perioperative therapy and D2 gastrectomy is currently considered the standard treatment option for locally advanced G/GEJ cancers (cT₃N₂₋₃M₀ and cT_{4a}N₊M₀, or cT_{4b}N_{any}M₀ after multidisciplinary discussion).¹”

- *Line 62: Checkmate-649 used a PD1 inhibitor so this sentence (i.e. describing efficacy of PD-1 and PD-L1 inhibitors in CheckMate-649) is not correct*

Response: We agree with the Reviewer. The statement was revised: “Immune checkpoint inhibitors that target programmed cell death protein 1 (PD-1) or programmed cell death ligand 1 (PD-L1) have shown promising survival benefit and manageable safety in first-line treatment of patients with G/GEJ cancers in the CheckMate-649,² GEMSTONE-101,³ and ORIENT-16 trials.⁴”

- *Needs to be more nuanced in the discussion of nab-paclitaxel and its place in the treatment of this malignancy as this is not a component of the standard of care perioperative regimens-*

Response: We thank the Reviewer for the comment. Nab-paclitaxel was selected in the present study for several reasons. First, there is evidence of the efficacy of paclitaxel in gastric cancer.⁵ Second, there is also evidence of the non-inferior efficacy of nab-paclitaxel vs. paclitaxel in previously treated advanced gastric cancer⁶ and significantly longer progression-free survival of S-1+nab-paclitaxel vs. SOX in the first line treatment of advanced gastric cancer.⁷ Third, nab-paclitaxel does not require premedication with corticosteroids, which might be more suitable in the context of the unsure impact of corticosteroids in patients receiving immunotherapy.^{8,9} Although there

is no compelling evidence that corticosteroids decrease the efficacy of immunotherapy,¹⁰ corticosteroids and immunotherapy influence the immune system in different ways,^{8,9} and it was considered prudent when designing the trial to avoid that potential confounder. We added some discussion about that.

- *Line 82: Gives the impression that durvalumab is added to neoadjuvant CRT based on CM577 – it is not, it is an adjuvant monotherapy.*

Response: We agree with the Reviewer. It was revised: “The CheckMate-577 study demonstrated the survival benefit of adjuvant nivolumab in patients with esophageal/GEJ cancer who received preoperative chemoradiotherapy.¹¹ However, the efficacy and safety of perioperative immunotherapy plus cCRT in locally advanced gastric cancer remain mostly unexplored.¹²”

Methods:

- *Unclear from this document whether any pre-treatment physiological exclusion criteria? E.g renal function assessment? Lung function assessment?*

Response: We thank the Reviewer. The study protocol was published previously.¹³ As stated in the protocol, the patients had to have adequate physiological functions, as indicated by absolute neutrophil count $\geq 1.5 \times 10^9/L$, platelets $\geq 100 \times 10^9/L$, hemoglobin ≥ 90 g/L, international normalized ratio/prothrombin time $\leq 1.5 \times$ the upper limit of normal (ULN), activated partial thromboplastin time $\leq 1.5 \times$ ULN, plasma total bilirubin $\leq 1.5 \times$ ULN, alanine transaminase $\leq 2.5 \times$ ULN, aspartate transaminase $\leq 2.5 \times$ ULN, alkaline phosphatase $\leq 2.5 \times$ ULN, creatinine $\leq 1 \times$ ULN, and albumin ≥ 30 g/L.¹³ The reference was provided and the protocol is freely available, and the detailed eligibility criteria are not listed in the manuscript due to the word limit.

- *These are mentioned in protocol so needs to more explicitly point to this as a references. Same goes for the staging investigations and physiological tests which were required pre-treatment?*

Response: We understand the Reviewer’s concerns. The physical examination, weight, ECOG performance status, vital signs, routine laboratory tests (blood routine,

blood chemistry, blood coagulation routine, urine routine, stool routine for occult blood, and thyroid function), 12-lead electrocardiogram (ECG) and echocardiography were assessed in all patients within 7 days before starting the study. Besides, tumor assessment by means of computed CT/MRI and EUS will be done at baseline (within 4 weeks before enrolment). Due to the word limit of the main text, it was not detailed in the Method, and the published study protocol is available open-access on PubMed Central.¹³

- The greatest concern, considering that this is a chemoradiotherapy study, is that the radiotherapy delivery is not described in sufficient detail and the quality control of RT is not clear. For example it is:

o Unclear what radiotherapy simulation was performed

o Unclear where RT delivered to – very scanty details

o Presumably delivered 5 fractions per week?

o What was the target volume?

♣ Entire stomach? What if GOJ – to what point then?

♣ Regional lymph nodes? Which?

♣ Margins used?

o How was target volume grown to CTV and PTV?

o On-treatment imaging used? Any immobilisation?

o What were the OARs? To what tolerance?

o What was the technique?

♣ 3D or 4D?

♣ Conformal or IMRT/VMAT?

o What happened if treatment interrupted? Catch up to Rx within normal time?

o Was there a standard RT protocol?

o Any QA used?

o Were patients reviewed whilst on-treatment? At what interval?

o How were adverse events defined as relating to radiotherapy rather than other components of the treatment?

Response: We thank the Reviewer for the comment. As per protocol, the radiotherapy was given according to local physicians' preferences, guidelines, experience, and tumor board discussions. Due to the word limit of the manuscript, the details of radiotherapy were not described in the main text. As for the interest of the reviewer about the detail and the quality control of radiotherapy in this study, we have attached the protocol of radiotherapy, including the selection of radiotherapy technology, positioning and simulation, definition of the targeted area, dose and regimen, normal structures and constraints, image-guided technology and quality assurance.

Regarding the interruption of radiotherapy, if the duration of interruption was within one week, the treatment would not be adjusted. Otherwise, the total treatment period, total dose and dose per treatment would maintain unadjusted and the catch-up would be done by additional radiation in one day or at weekend. Patients visited the attending doctor weekly and routine laboratory tests were performed accordingly.

Indeed, it is difficult to distinguish the cause of adverse events when combined therapy was applied due to similar symptoms. In many clinical trials on radiotherapy or chemotherapy combined with immunotherapy, the AEs are not subdivided into specific drugs but are counted as the total AEs in the combined treatment mode. For example, CheckMate 649 and RTOG 9904.^{2,14} It was briefly described in the Methods. Hence, the results and table of radiotherapy-related AEs were revised to concurrent chemoradiotherapy-related adverse events after careful consideration.

- The same can be said for surgery. Please describe in detail which surgical was performed?

Response: We thank the Reviewer for this fair comment. Similar with the details of radiotherapy, due to the word limit of the manuscript, the details of surgery were not

described in the main text and the protocol of surgery was attached as Supplementary Materials.

- *Please also provide the following information:*

- *How did they choose their panel of biomarkers to evaluate?*

Response: We thank the Reviewer. The biomarker panel was decided by the trial committee through discussion when designing the trial, based on guidelines^{1,15} and experience. This panel of 10 biomarkers was selected based on the efficacy of immunotherapy in solid tumors, including biomarkers for T cells (CD3, CD4, and CD8), Tregs (FoxP3), B cells (CD20), NK cells (CD56), macrophages (CD68, CD163), and immune escape (PD-1/L1). For example, CD8 T cell tumor infiltration was positively correlated with the immunotherapy response in melanoma and uroepithelial tumors.^{16,17} CD4⁺FoxP3⁺ Tregs are significantly elevated in gastric cancer responsive to immunotherapy.¹⁸ CD20 B cells are associated with the response to immunotherapy in melanoma.¹⁹ CD56 NK cells are important immune cells that play a critical role in the early anti-tumor immune response by directly killing tumor cells.²⁰ Immunosuppressive CD68⁺CD163⁺ M2 macrophages are negatively associated with the efficacy of immunotherapy in ovarian cancer.²¹ Of note, those biomarkers are for exploratory purposes only. It was clarified in the Methods.

- *How did they decide which component of therapy to attribute adverse events to?*

Response: We thank the Reviewer for the comment. Judgment and grading of the AEs were based on the time of occurrence of the AEs, clinical manifestations, and relevant test and examination results. If necessary, multidisciplinary discussions were held with relevant departments such as imaging, immunology, gastroenterology, and respiratory. For example, one of the participants had grade III aminotransferase elevation and grade IV bilirubin elevation. The patient's test results differed from the common manifestations of chemotherapy-related liver damage, and conventional liver-protective and enzyme-lowering therapy was ineffective. Therefore, it was considered

that these AEs were immunotherapy-related after discussion, and the patient's liver function improved after treatment according to the CSCO Guidelines for the Management of Toxicity Associated with Immune Checkpoint Inhibitors. Still, many AEs can occur with combined treatment, and it is true that it is difficult to distinguish which specific part is causative, such as nausea and vomiting, and other gastrointestinal reactions, which can be caused by both radiotherapy and chemotherapy, and their clinical manifestations and management are similar. For such AEs, there was no specific distinction regarding which part of the treatment was causative, but conventional clinical symptomatic management was carried out. In many clinical trials on radiotherapy or chemotherapy combined with immunotherapy, the AEs are not subdivided into specific drugs but are counted as the total AEs in the combined treatment mode. For example, CheckMate 649 and RTOG 9904.^{2,14} It was briefly described in the Methods. Hence, the results and table of radiotherapy-related AEs were revised to concurrent chemoradiotherapy-related adverse events after careful consideration.

- Was there an IDMC to decide whether to progress from first to second stage of trial design?

Response: We thank the Reviewer. There was no IDMC and a Simon 2-stage design was employed in this study. An ORR $\leq 15\%$ for an anti-PD-1 antibody was deemed unacceptable, but an ORR $\geq 35\%$ was considered to warrant additional study. Using a power of 80% and $\alpha=0.05$, a minimum of nine patients were needed in stage I, and if a pCR was confirmed in one or two of them, an additional 25 patients were enrolled in stage II. It was clarified in the Methods.

- Any adjustment for multiple significance testing in the biomarker analysis groups?

Response: The Reviewer raises a good point. We agree that taking into account multiple testing is methodologically important. The multiple testing was taken into account as suggested and repeated measures were tested using Friedman's non-parametric test and the Bonferroni correction. We found that only CD56⁺ and CD56^{dim}

were positively correlated with pCR with a Bonferroni adjusted P-value of 0.024 and 0.012 respectively. Still, we have to stress that the biomarker analysis was purely exploratory. We cannot reach any firm conclusion about the biomarkers because of the small sample size. In addition, Bonferroni correction test might be overly conservative and lead to irrelevant null hypotheses. Presenting biomarkers with only unadjusted P value < 0.05 are hypothesis-generating and will be examined in future studies.

- *Was there an a priori stats analysis plan?*

Response: Yes, the analysis plan was designed before conducting the trial and the preplanned statistical analysis plan was attached as Supplementary Materials. It was clarified in the Methods.

- *Why did they drop the S-1 during CRT?*

Response: We thank the Reviewer for the question. Considering the use of concurrent treatment with radiotherapy, chemotherapy and immunotherapy might raise the issue of tolerability, we only administered single chemotherapy regimen during CRT. Most studies of cCRT regimens for gastric cancer used S-1. Nab-paclitaxel has a synergistic effect with immunotherapy.^{22,23} In addition, nab-paclitaxel has a synergistic effect with radiotherapy in lung cancer, pancreatic cancer, and head and neck squamous cancer, among others.²⁴⁻²⁷ Therefore, in this study, nab-paclitaxel was selected in the cCRT phase in order to pursue a better tumor control rate with optimal clinical benefit for patients. It was clarified in the Discussion.

Results

- *Line 98: 'a total of 34 patients were included in the analysis' – do they mean enrolled?*

Makes it sound like some were treated but not analysed

Response: We meant enroll. It was corrected.

- *Line 105: all patients completed neoadjuvant therapy and underwent surgery*

o Not actually true

Response: The Reviewer is right. We meant received. It was corrected.

- *Patient group:*

o No known HER2 status

Response: We thank the Reviewer. We added the HER2 status in Table 1.

- *Radiotherapy-related adverse events in 65%; 24% with grade 3 and 1 with grade 4.*

- *Two patients had RT discontinuation*

o Supp table 1 indicates one due to a G1 AE and one due to a G3 AE – need extra info on those

o At what point was treatment discontinued? What dose had these patients received?

What were the AE? Seems strange that a G1 AE led to discontinuation

Response: We thank the Reviewer for the comment and sorry for the mistake. Indeed, two patients had RT discontinuation due to cCRT-related adverse events: One patient due to G3 nausea/vomiting at the dose of 36Gy/20f and one due to G3 nausea/vomiting and myelosuppression at the dose of 34.2Gy/19f. The results and tables were updated.

- *All RT delivered within planned treatment time? Any interruptions or catch up required? How was catch up done?*

Response: We thank the Reviewer for this fair comment. As stated above, two patients discontinued radiotherapy due to adverse events, and all other patients complete the radiotherapy per protocol. The catch-up and management of radiotherapy interruption was described in the response to the comment on Methods of radiotherapy: Regarding the interruption of radiotherapy, if the duration of interruption was within one week, the treatment would not be adjusted. Otherwise, the total treatment period, total dose and dose per treatment would maintain unadjusted and the catch-up would be done by additional radiation in one day or at weekend. Patients visited the attending doctor weekly and routine laboratory tests were performed accordingly.

- Says all patients completed neoadjuvant therapy and underwent surgery
- o BUT – two patients discontinued radiotherapy due to rAEs, so this can't be right

Response: The Reviewer is right. We meant received. It was corrected.

- Toxicity from RT:
 - o Rates of fatigue seem remarkably low
 - ♣ Only 1 grade 2 fatigue – 67% of CROSS had fatigue of at least grade 1 in severity
 - o Pneumonia quoted but concern is pneumonitis – was this in the GOJ group?
 - o No skin tox?
 - o What is 'gastrointestinal discomfort'?
 - o Would want specific yes/no to usual list of toxicities expected e.g. as per SCOPE/CROSS GOJ cases

Response: We thank the Reviewer for the comment. After double-checking the data, it is indeed only one case of grade 2 fatigue. It might be attributed to the relatively small sample size. The observation and recording of related adverse reactions will be more standardized in future phase III clinical trials.

The Reviewer is right. "Pneumonia" was an error in translation. In one patient, it was immune-related pneumonitis; in another patient, it was infectious pneumonia after surgery. It was clarified in the manuscript. Yes, they were both in the GOJ group.

No skin toxicity was recorded. In fact, all AEs that were recorded during the treatments were indicated in Table 3. Therefore, AEs that are not indicated in Table 3 had an occurrence of 0. It was clarified as a footnote to Table 3.

We are sorry for the confusion caused by inaccurate expression. Indeed, gastrointestinal means nausea and vomiting. Hence, we proofread through the whole manuscript.

- General toxicity:
 - o 9% rate of fever in the neoadjuvant period ◊ compared with 80% myelosuppression?
- Seems very low but unclear when bloods were taken and how often

o One patient had intestinal obstruction? Did they then not receive adjuvant rx?

o What is gastrointestinal discomfort?

Response: We thank the Reviewer for the comment. Routine blood tests and blood biochemistry were reviewed weekly. Tumor biomarkers and thyroid function were reviewed preoperatively at weeks 3 and 8 of neoadjuvant therapy and postoperatively every 3 weeks. It was clarified in the Methods. When the number of white blood cells or neutrophils decreases, the frequency of blood test increased. Prophylactic anti-infective therapy was given to patients with grade IV leukopenia with fever, which might contribute to a low incidence of fever. Patients with postoperative intestinal obstruction improved after symptomatic treatment and completed follow-up adjuvant therapy. Gastrointestinal discomfort means nausea/vomiting.

- Unclear long-term follow-up:

o Any post-op cardiac tox? Lung tox?

o Adverse events only studied up to 90 days post-completion of last bit of IO

♣ Treatment ended at 13 weeks post-op \diamond so 3 + 3 months. Should have captured pneumonitis but not long enough for cardiac sequelae so needs commenting on

Response: We thank the Reviewer. At the time of revision, no serious cardiac or pulmonary AEs were recorded.

- Need to better report surgical complications using Clavien-Dindo grading

Response: We thank the Reviewer. They are now provided in Supplementary Table 3.

- Two patients discontinued RT:

o Why? What was the rrAE that led to them discontinuing?

o What dose did they receive by the time they discontinued?

- Anything about functional hyposplenism?

o Did they cover with abx for this?

Response: We thank the Reviewer for the comment. As stated above, two patients

discontinued radiotherapy due to adverse events: ne patient due to grade 3 nausea/vomiting at the dose of 36Gy/20f and one due to grade 3 nausea/vomiting and myelosuppression at the dose of 34.2Gy/19f. The two patients refused to receive further radiation. It was added in the Results. No functional hyposplenism was observed.

Translational research,

The investigators have examined a few biomarkers but it's largely descriptive. As pre and post samples are available, it would be interesting to look at the change in biomarker expression on a per patient level.

Response: We thank the Reviewer for the comment. Indeed, the biomarkers are mostly descriptive because these analyses were exploratory. The sample size does not allow for any strong conclusion on biomarkers. Still, we added the changes in the biomarkers as Supplementary Figure 5.

Discussion

Notes that CROSS found a pCR of 29.2% and POET of 15.6% in dCRT groups

o CROSS actually found a pCR of 23% in patients with adenocarcinoma (i.e. of relevance to this description) so could be better written to help uninformed reader

Response: The Reviewer is right. It was clarified in the manuscript: “In the neoadjuvant setting, the CROSS and POET trials investigated the feasibility of cCRT in patients with esophagus or GEJ cancer, with pCR rates of 29.2% (or 23% in patients with adenocarcinoma) and 15.6%, respectively.^{28,29}”

Would be helpful to correlate more to existing literature with respect to concurrent immunotherapy with CRT, as is an important question in the field. Would also help to explain significance of biomarker analyses. Ideally they should mention TOPGEAR but also need to discuss FLOT/FLOT-IO etc

Response: We thank the Reviewer for the comment. Indeed, the biomarkers are mostly descriptive because these analyses were exploratory. The sample size does not allow for any strong conclusion on biomarkers. It was instead added as a Limitation.

There are few studies of CRT + ICI. The manuscript already discussed Zhu et al.³⁰ and the Neo-PLANET trial.³¹ We did not find other relevant research articles on CRT + ICI for the time being. We agreed with the reviewer that the TOPGEAR trial should be discussed, but TOPGEAR has not yet published its results, and only the protocol is available.³² Still, the interim results suggest that preoperative CRT is safe for the vast majority of patients without additional treatment toxicity or surgical morbidity.³³ The benefit and risk of intensive chemotherapy regimen in TOPGEAR study (epirubicin, cisplatin and fluorouracil) would be properly discussed when the full results were disclosed, along with the results of ongoing clinical trials that use the combination of FLOT and immunotherapy (e.g. MATTERHORN study, NCT04592913; KEYNOTE-585 study, NCT03221426). We added some discussion about them.

Further study

o Worth pointing to the potential benefits of different fractionation regimes and how perhaps intensified systemic with shorter course RT might work etc

Response: We thank the Reviewer. We added to the Discussion, “In the present study, radiotherapy was given according to local physicians’ preferences, according to guidelines, experience, and tumor board discussions. Future studies could look into specific radiotherapy regimens for preoperative sintilimab in combination with cCRT. Different doses and fractionation regimens could yield different results and should be explored in the future.”

Reviewer #2

1. What is the rationale for selecting the preoperative chemotherapy regimen? The recommended regimens were usually consisting of 5-Fu and platins in China.

Response: We thank the Reviewer for the comment. As stated in the protocol:¹³ “In addition, in Japan, D2 gastrectomy plus adjuvant S-1 is the standard treatment for locally advanced gastric cancer [18, 19], while nab-paclitaxel (nab-PTX) is an approved second-line gastric cancer treatment [23, 24]. The combination of S-1 with nab-PTX has proven to be an effective and safe first-line regimen in clinical [25, 26]. Furthermore, the interim analysis from the JACCRO GC-07 trial has demonstrated that the postoperative S-1 plus docetaxel is effective with few safety concerns in patients with stage III gastric cancer [25] and similar benefits were also noted with S-1 plus nab-PTX in untreated patients with metastatic gastric cancer [26]”. The rationale for selecting the preoperative chemotherapy regimen was clarified in the Introduction.

2. What is the rationale for setting reference pCR rate as 15%? In Cross trial the pCR rate was 29%, and in the trials conducted by Ajani, the pCR rates were reported 19.5%, 26% and 30% in gastric cancer after nCRT respectively (JCO 2005, JCO 2006, JCO 2004).

Response: We thank the Reviewer for the comment. These numbers were discussed and selected based on the differences in patient characteristics. In the CROSS study,²⁸ the patients were T₂₋₃N₀₋₁M₀ stage, and the pCR was 23% in patients with adenocarcinoma, lower than in squamous carcinoma. In the RTOG 9904 study,¹⁴ the patients were T₂₋₃N₀₋₁M₀ or T₁N₁M₀, and the pCR was 23%. Therefore, these two studies included patients with lower-stage cancer than in the present study. On the other hand, in the RESOLVE study,³⁴ patients were more advanced than in the present study, with cT_{4a}N₊M₀ or cT_{4b}N_xM₀ without radiotherapy, with a pCR of only 5.6%.

3. In trials studying ICIs in perioperative therapy of gastric cancer, the cycles of ICIs were much more than it in this trial which only included three cycles of sintilimab. For

example, in Keynote-585, 3 cycles of pembrolizumab with chemo followed by 11 cycles of pembrolizumab alone, in DANTE trial, 4 cycles of atezolizumab with chemo followed by 8 cycles of atezolizumab alone. Please discuss the difference.

Response: We agree with the Reviewer, but when the present trial was designed, not many trials were available for guidance. Following regular perioperative treatment strategies, we designed a 3-cycle treatment before and after the surgery, respectively in this study. NCT05687357 is an ongoing large-scale study we conducted based on the results of SHARED study that use immunotherapy for 1 year in the neoadjuvant treatment for G/GEJ adenocarcinoma.

4.The aim of this study was to evaluate the efficacy and safety of perioperative therapy, but the primary endpoint was pCR which only indicates the efficacy of preoperative therapy. For perioperative therapy, DFS is more reasonable. I can understand the difficult to conduct such a trial with survival outcome as primary endpoint, some discussion about this topic will be preferred.

Response: We thank the Reviewer for the comment. This study was a preliminary exploratory phase II trial, and selecting the pCR as the primary endpoint is common in such trials of solid tumors as it provides an answer more rapidly than survival.³⁵⁻³⁷ In the RTOG 9904 trial, the patients with pCR achieved better outcomes than the non-pCR patients and pCR was suggested as a prognostic surrogate,^{14,36} similar to the results in this study that patients achieving pCR had significantly longer DFS than those did not (20.9 vs. 11.1 months, P=0.0285). Future studies will have a longer follow-up. For now, it was added as a Limitation.

5.Could the authors provide more details of irAEs?

Response: We thank the Reviewer. The data of irAEs were provided as Supplementary Table 2.

6.There were 6 patients with Bormann IV tumor, how about treatment responses in these tumors?

Response: We thank the Reviewer. Among the six participants with Bormann IV tumors, the MPR was 83.3% (5/6), the pCR rate was 16.7% (1/6), the rate of R0 resection was 100% (6/6), one was ypT₀, one was ypT₂, four were ypT₃, three were ypN₀, two were ypN₂, and one was ypN_{3a}. It was added to the Results.

7. One patient received proximal gastrectomy, this type of gastrectomy was controversial in locally advanced gastric cancer and with little evidence in post-neoadjuvant therapy patients. The KLASS05 trial, a randomized trial comparing proximal gastrectomy (PG) with total gastrectomy (TG), found more advanced pN stage in TG than in PG, which indicated the proximal gastrectomy may have limitation in lymph node dissection. So, proximal gastrectomy should be performed with caution.

Response: We thank the Reviewer. Indeed, one patient underwent proximal gastrectomy. The tumor was in the AGJ and involved the lower esophagus. The thoracic surgeon was consulted during the operation and assessed that it would be difficult to perform a total gastrectomy with an esophagojejunostomy. Therefore, proximal gastrectomy and esophagogastrostomy were performed.

8. To my knowledge TLS was more likely to present in the border of submucosa and muscle layer, endoscopy biopsy was usually superficial and may underestimate the abundance of TLS. Among the patient with TLS in baseline, pCR rate was 40% (2/5) similar to 38.2% in whole cohort. MPR rate was 80% similar to 79.2% in whole cohort too. Please revise the discussion more precisely.

Response: We thank the Reviewer for the comment. Indeed, the biomarkers are mostly descriptive because these analyses were exploratory. The sample size does not allow for any strong conclusion on biomarkers. It was instead added as a Limitation.

Nevertheless, TLSs are more abundant in the invasive margin or the stroma than in the core of the tumors. Therefore, the whole sectioning of a tumor block with the most stromal infiltration is a reliable approach to detecting TLSs.³⁸

Therefore, as the Reviewer mentioned, tumor biopsy samples are taken from only a small portion of tissue, which is not very favorable for TLS detection and may be

missed. It may also be one of the reasons why we detected that the proportion of resections in the TLS group was comparable to the whole cohort, and TLS did not become a predictive biomarker for the efficacy of immunotherapy combined with radiotherapy.

Hence, we revised the discussion: “Among the patient with TLS at baseline, the pCR rate was 40% (2/5), similar to 38.2% in the whole cohort. The MPR rate was 80%, also similar to 79.2% in the whole cohort. It is possible that endoscopy biopsy does not cover large tumor stromal areas rich in TLS structure, and the number of TLSs will be underestimated.³⁸”

9. In Figure 3A, the density of CD3+ T cell was dramatically different between whole tumor and stroma in pCR samples, I am wondering the reason why?

Response: We thank the Reviewer for the comment. After double-checking the data, we confirmed that it was correct. In the literature, T-cells (CD4 and CD8) are significantly more abundant in tumors than in stroma.³⁹ Since the T cells need to infiltrate into the tumor to exert their tumor-killing effect, it is reasonable that CD3+, an important marker of T-cells, is found significantly more in the tumor stroma than in the tumor, especially in treatment-responsive better tumors. Nevertheless, the biomarkers are mostly descriptive because these analyses were exploratory. The sample size does not allow for any strong conclusion on biomarkers. It was instead added as a Limitation.

Reviewer #3

1) Endpoint definitions of time-to-event endpoints need to include the definition that was used for censoring for each endpoint (DFS, EFS, and OS) separately.

Response: We thank the Reviewer for the comment. DFS was defined as the time from surgery to postoperative recurrence or death from any cause, whichever occurred first. DFS was censored on the last tumor assessment date for patients still alive and without recurrence. EFS was the time from enrolment to recurrence or death from any cause. EFS was censored on the last tumor assessment date for patients still alive and without recurrence. OS was the time from enrolment to death from any cause. OS was censored on the last date known to be alive for patients without documentation of death. It was clarified in the Methods.

2) Table 2 presents many endpoints that are binary with the count, proportion, and a confidence interval. The primary endpoint is clearly presented with a label for the type of confidence interval and the level. Please add to the methods and the table the method and alpha for all of the other confidence intervals presented.

Response: We thank the Reviewer. The confidence intervals of pCR, MPR, R0 resection, pathological T stages, and pathological N stages were estimated by the Clopper-Pearson exact method. The alpha levels for confidence intervals of pCR and other proportions were 90% and 95%, respectively. We added these details in the Methods and Table 2.

3) The methods for analysis of the pCR outcome do not include enough detail to produce Supplement Figure 2. Please add the methods for the confidence intervals included. Additionally, the figure would benefit with presentation with less precision of the estimates. Considering the sample size, 2 significant digits is sufficient. Additionally, the title of the graph portion of the figure should include that it is proportion with pCR.

Response: We thank the Reviewer for the comment. We revised Supplementary Figure 2, as below.

4) The analysis of Disease-Free-Survival presented in supplement Figure 3 presents Kaplan-Meier Figures. There is a p-value presented under a hazard ratio on a figure estimated with non-parametric Kaplan-Meier methods. It is unclear if the p-value represents a test using parametric or non-parametric methods. Please clarify what the p-value is testing. Additionally, consider removing the hazard ratios from the figure due to the mix of methods in each figure.

Response: We thank the Reviewer for the comment. We revised Supplementary

Figure 3, as below.

References

- 1 Wang, F.-H. *et al.* The Chinese Society of Clinical Oncology (CSCO): Clinical guidelines for the diagnosis and treatment of gastric cancer, 2021. *Cancer Commun (Lond)* **41**, 747-795, doi:10.1002/cac2.12193 (2021).
- 2 Janjigian, Y. Y. *et al.* First-line nivolumab plus chemotherapy versus chemotherapy alone for advanced gastric, gastro-oesophageal junction, and oesophageal adenocarcinoma (CheckMate 649): a randomised, open-label, phase 3 trial. *The Lancet* **398**, 27-40, doi:10.1016/s0140-6736(21)00797-2 (2021).
- 3 Shen, L. *et al.* 1445P - CS1001, an anti-PD-L1 antibody, combined with standard of care (SOC) chemotherapy for first line (1L) advanced GC/GEJ and ESCC: Preliminary results from 2 phase Ib cohorts of CS1001-101 study. *Ann Oncol* **31**, S841-S873, doi:10.1016/annonc/annonc284 (2020).
- 4 Xu, J. *et al.* LBA53 Sintilimab plus chemotherapy (chemo) versus chemo as first-line treatment for advanced gastric or gastroesophageal junction (G/GEJ) adenocarcinoma (ORIENT-16): First results of a randomized, double-blind, phase III study. *Annals of Oncology* **32**, S1331, doi:10.1016/j.annonc.2021.08.2133 (2021).
- 5 Gadgeel, S. M. *et al.* Phase II study of paclitaxel and carboplatin in patients with advanced gastric cancer. *Am J Clin Oncol* **26**, 37-41, doi:10.1097/00000421-200302000-00008 (2003).
- 6 Shitara, K. *et al.* Nab-paclitaxel versus solvent-based paclitaxel in patients with previously treated advanced gastric cancer (ABSOLUTE): an open-label, randomised, non-inferiority, phase 3 trial. *Lancet Gastroenterol Hepatol* **2**, 277-287, doi:10.1016/S2468-1253(16)30219-9 (2017).
- 7 Dai, Y. H. *et al.* Nab-paclitaxel plus S-1 versus oxaliplatin plus S-1 as first-line treatment in advanced gastric cancer: results of a multicenter, randomized, phase III trial (GAPSO study). *Ther Adv Med Oncol* **14**, 17588359221118020, doi:10.1177/17588359221118020 (2022).
- 8 Aldea, M. *et al.* How to manage patients with corticosteroids in oncology in the era of immunotherapy? *Eur J Cancer* **141**, 239-251, doi:10.1016/j.ejca.2020.09.032 (2020).
- 9 Janowitz, T., Kleeman, S. & Vonderheide, R. H. Reconsidering Dexamethasone for Antiemesis when Combining Chemotherapy and Immunotherapy. *Oncologist* **26**, 269-273, doi:10.1002/onco.13680 (2021).
- 10 Nelson, B. E. *et al.* Corticosteroid use and its impact on the efficacy of immunotherapy in multiple tumor types. *J Clin Oncol* **39**, doi:10.1200/JCO.2021.39.15_suppl.e14583 (2021).
- 11 Kelly, R. J. *et al.* Adjuvant Nivolumab in Resected Esophageal or Gastroesophageal Junction Cancer. *N Engl J Med* **384**, 1191-1203, doi:10.1056/NEJMoa2032125 (2021).
- 12 Li, S. *et al.* Neoadjuvant therapy with immune checkpoint blockade, antiangiogenesis, and chemotherapy for locally advanced gastric cancer. *Nat Commun* **14**, 8, doi:10.1038/s41467-022-35431-x (2023).
- 13 Wei, J. *et al.* Efficacy and Safety of Sintilimab in Combination with Concurrent Chemoradiotherapy for Locally Advanced Gastric or Gastroesophageal Junction (GEJ) Adenocarcinoma (SHARED): Study Protocol of a Prospective, Multi-Center, Single-Arm Phase 2 Trial. *Cancer Manag Res* **14**, 2007-2015, doi:10.2147/CMAR.S355687 (2022).
- 14 Ajani, J. A. *et al.* Phase II trial of preoperative chemoradiation in patients with localized gastric adenocarcinoma (RTOG 9904): quality of combined modality therapy and pathologic response. *J Clin Oncol* **24**, 3953-3958, doi:10.1200/JCO.2006.06.4840 (2006).

- 15 Ajani, J. A. *et al.* Gastric Cancer, Version 2.2022, NCCN Clinical Practice Guidelines in
Oncology. *J Natl Compr Canc Netw* **20**, 167-192, doi:10.6004/jnccn.2022.0008 (2022).
- 16 Tumei, P. C. *et al.* PD-1 blockade induces responses by inhibiting adaptive immune resistance.
Nature **515**, 568-571, doi:10.1038/nature13954 (2014).
- 17 Rosenberg, J. E. *et al.* Atezolizumab in patients with locally advanced and metastatic urothelial
carcinoma who have progressed following treatment with platinum-based chemotherapy: a
single-arm, multicentre, phase 2 trial. *Lancet* **387**, 1909-1920, doi:10.1016/S0140-
6736(16)00561-4 (2016).
- 18 Kamada, T. *et al.* PD-1(+) regulatory T cells amplified by PD-1 blockade promote
hyperprogression of cancer. *Proc Natl Acad Sci U S A* **116**, 9999-10008,
doi:10.1073/pnas.1822001116 (2019).
- 19 Helmink, B. A. *et al.* B cells and tertiary lymphoid structures promote immunotherapy response.
Nature **577**, 549-555, doi:10.1038/s41586-019-1922-8 (2020).
- 20 Watkins-Schulz, R. *et al.* A microparticle platform for STING-targeted immunotherapy
enhances natural killer cell- and CD8(+) T cell-mediated anti-tumor immunity. *Biomaterials*
205, 94-105, doi:10.1016/j.biomaterials.2019.03.011 (2019).
- 21 Hensler, M. *et al.* M2-like macrophages dictate clinically relevant immunosuppression in
metastatic ovarian cancer. *J Immunother Cancer* **8**, doi:10.1136/jitc-2020-000979 (2020).
- 22 Chen, L. *et al.* Famitinib with Camrelizumab and Nab-Paclitaxel for Advanced
Immunomodulatory Triple-Negative Breast Cancer (FUTURE-C-Plus): An Open-Label,
Single-Arm, Phase II Trial. *Clin Cancer Res* **28**, 2807-2817, doi:10.1158/1078-0432.CCR-21-
4313 (2022).
- 23 Zhang, J. *et al.* Synergistic effects of nab-PTX and anti-PD-1 antibody combination against lung
cancer by regulating the Pi3K/AKT pathway through the Serpincl gene. *Front Oncol* **12**,
933646, doi:10.3389/fonc.2022.933646 (2022).
- 24 Takano, N. *et al.* Phase II study of chemoradiotherapy combined with gemcitabine plus nab-
paclitaxel for unresectable locally advanced pancreatic ductal adenocarcinoma (NUPAT 05
Trial): study protocol for a single arm phase II study. *Nagoya J Med Sci* **81**, 233-239,
doi:10.18999/nagjms.81.2.233 (2019).
- 25 Tanaka, H. *et al.* A Phase I/II Study of Biweekly Carboplatin and Nab-paclitaxel With
Concurrent Radiotherapy for Patients With Locally Advanced Unresectable Stage III Non-
small-cell Lung Cancer. *Clin Lung Cancer* **22**, 42-48, doi:10.1016/j.clc.2020.09.016 (2021).
- 26 Wu, K. *et al.* A phase II study of concurrent nab-paclitaxel/carboplatin combined with thoracic
radiotherapy in locally advanced squamous cell lung cancer. *J Thorac Dis* **11**, 4529-4537,
doi:10.21037/jtd.2019.10.81 (2019).
- 27 Yamada, S. *et al.* Phase I study of chemoradiotherapy using gemcitabine plus nab-paclitaxel for
unresectable locally advanced pancreatic cancer. *Cancer Chemother Pharmacol* **81**, 815-821,
doi:10.1007/s00280-018-3554-3 (2018).
- 28 Shapiro, J. *et al.* Neoadjuvant chemoradiotherapy plus surgery versus surgery alone for
oesophageal or junctional cancer (CROSS): long-term results of a randomised controlled trial.
The Lancet Oncology **16**, 1090-1098, doi:10.1016/s1470-2045(15)00040-6 (2015).
- 29 Stahl, M. *et al.* Preoperative chemotherapy versus chemoradiotherapy in locally advanced
adenocarcinomas of the oesophagogastric junction (POET): Long-term results of a controlled
randomised trial. *Eur J Cancer* **81**, 183-190, doi:10.1016/j.ejca.2017.04.027 (2017).

- 30 Zhu, M. *et al.* Pembrolizumab in Combination with Neoadjuvant Chemoradiotherapy for Patients with Resectable Adenocarcinoma of the Gastroesophageal Junction. *Clin Cancer Res* **28**, 3021-3031, doi:10.1158/1078-0432.CCR-22-0413 (2022).
- 31 Tang, Z. *et al.* The Neo-PLANET phase II trial of neoadjuvant camrelizumab plus concurrent chemoradiotherapy in locally advanced adenocarcinoma of stomach or gastroesophageal junction. *Nat Commun* **13**, 6807, doi:10.1038/s41467-022-34403-5 (2022).
- 32 Leong, T. *et al.* TOPGEAR: a randomised phase III trial of perioperative ECF chemotherapy versus preoperative chemoradiation plus perioperative ECF chemotherapy for resectable gastric cancer (an international, intergroup trial of the AGITG/TROG/EORTC/NCIC CTG). *BMC Cancer* **15**, 532, doi:10.1186/s12885-015-1529-x (2015).
- 33 Leong, T. *et al.* TOPGEAR: A Randomized, Phase III Trial of Perioperative ECF Chemotherapy with or Without Preoperative Chemoradiation for Resectable Gastric Cancer: Interim Results from an International, Intergroup Trial of the AGITG, TROG, EORTC and CCTG. *Ann Surg Oncol* **24**, 2252-2258, doi:10.1245/s10434-017-5830-6 (2017).
- 34 Zhang, X. *et al.* Perioperative or postoperative adjuvant oxaliplatin with S-1 versus adjuvant oxaliplatin with capecitabine in patients with locally advanced gastric or gastro-oesophageal junction adenocarcinoma undergoing D2 gastrectomy (RESOLVE): an open-label, superiority and non-inferiority, phase 3 randomised controlled trial. *Lancet Oncol* **22**, 1081-1092, doi:10.1016/S1470-2045(21)00297-7 (2021).
- 35 Jiang, H. *et al.* Efficacy and safety of neoadjuvant sintilimab, oxaliplatin and capecitabine in patients with locally advanced, resectable gastric or gastroesophageal junction adenocarcinoma: early results of a phase 2 study. *J Immunother Cancer* **10**, doi:10.1136/jitc-2021-003635 (2022).
- 36 Ren, S. *et al.* A narrative review of primary research endpoints of neoadjuvant therapy for lung cancer: past, present and future. *Transl Lung Cancer Res* **10**, 3264-3275, doi:10.21037/tlcr-21-259 (2021).
- 37 Conforti, F. *et al.* Evaluation of pathological complete response as surrogate endpoint in neoadjuvant randomised clinical trials of early stage breast cancer: systematic review and meta-analysis. *BMJ* **375**, e066381, doi:10.1136/bmj-2021-066381 (2021).
- 38 Sautes-Fridman, C., Petitprez, F., Calderaro, J. & Fridman, W. H. Tertiary lymphoid structures in the era of cancer immunotherapy. *Nat Rev Cancer* **19**, 307-325, doi:10.1038/s41568-019-0144-6 (2019).
- 39 van Dijk, N. *et al.* The Tumor Immune Landscape and Architecture of Tertiary Lymphoid Structures in Urothelial Cancer. *Front Immunol* **12**, 793964, doi:10.3389/fimmu.2021.793964 (2021).

REVIEWER COMMENTS

Reviewer #1 (Remarks to the Author):

Page 1: looks fine to me

Page 2: I am still not really sure from the manuscript what the physiological exclusion criteria were (e.g. the authors state they are doing echocardiography but I'm not sure what the cut-offs were, whilst I also can't see any mention of lung function testing). I think these should be explicitly mentioned in the main manuscript.

Page 3/4: I think the 'radiotherapy was given according to local physicians' preferences...' and then the attachment of a radiotherapy protocol seem a bit conflicting; i.e. the reader is not sure what was the local physician's preference? Either way, I just don't think a manuscript focussed on radiotherapy should only have one line about how the radiotherapy was given and as a minimum I think they should outline (1) what their simulation process was, (2) what dose was administered (including Gy per fraction (which is to be fair given in the first para in procedures) and fractions per week) and broadly to what target volume (including the point to which the dose was prescribed), (3) which technique was used to deliver the radiation, (4) which particle was delivered (I assume photon) and of what energy (I assume 6 or 10MV) and (5) whether there were any radiotherapy quality assurance procedures. I suspect most rad oncs would also want at least a mention of whether they used protocols relating to gastric emptying pre-Rx given that it's hard to review tox without this info. Also, if the RT wasn't standardised or there wasn't QA then this should be explicitly outlined as a limitation in the discussion.

Page 6/7: It seems to be suggested that they should not adjust for multiple significance testing so as to be 'hypothesis generating', whereas obviously they are just then finding things by chance instead. It looks from the tracks like they added in something about repeated measures corrections to the stats section and then deleted it again. The biomarker figure should only be reported with p-values derived from correction for multiple significance testing.

Page 9: I think the fatigue data are so far from what would be expected that they need to highlight it in limitations and note that their tox reporting may well be underplaying the true toxicity.

Page 10: I think they should report in the manuscript itself that they used prophylactic antibiotics for those who had G4 leucopenia.

Page 13: worth stating explicitly that the CROSS stats we asked them to reference refer to adenocarcinomas

Reviewer #2 (Remarks to the Author):

Thanks for the reply from the authors, most of the responses answered my concerns very well. However, the reason for choosing a pCR rate of 15% as a reference was not convincing. Tumor stage was a reason that led to a higher pCR rate in Ajani's studies, but the pCR rate was 16% in the FLOT arm of the FLOT4 trial. The key point is what the standard of care is that this study is being compared to. Please provide further information for this question, "These numbers were discussed and selected based on the differences in patient characteristics" was not a good answer.

Reviewer #3 (Remarks to the Author):

The authors have addressed my concerns with this revision.

Reviewer #1

Page 1: looks fine to me

Response: Thanks for the comments. We have modified the manuscript to address your concerns about our manuscript below. Please kindly review.

Page 2: I am still not really sure from the manuscript what the physiological exclusion criteria were (e.g. the authors state they are doing echocardiography but I'm not sure what the cut-offs were, whilst I also can't see any mention of lung function testing). I think these should be explicitly mentioned in the main manuscript.

Response: We appreciate the reviewer's thoughtful question regarding the physiological exclusion criteria for our study, specifically related to echocardiography and lung function testing.

In our study, exclusion according to the results of echocardiography were made by the investigators according to widely accepted clinical standards. We agree with the reviewer that these criteria should be clearly stated to ensure the integrity and reproducibility of the study. However, it's worth noting that it is quite uncommon to provide such detailed physiological exclusion criteria within the main text of manuscripts published in **Nature Communications** [1, 2]. To substantiate this, we can refer to a number of recently published studies that also do not explicitly detail such physiological criteria within their main text [3-5]. These standards and parameters, while essential to the recruitment process, are often seen as part of the standard operating procedures in clinical trials. Still, we added the detailed physiological exclusion criteria in the revised manuscript, which were mostly judged by the investigators. Please kindly review.

Page 3/4: I think the 'radiotherapy was given according to local physicians' preferences...' and then the attachment of a radiotherapy protocol seem a bit conflicting; i.e. the reader is not sure what was the local physician's preference? Either way, I just don't think a manuscript focussed on radiotherapy should only have one line about how the radiotherapy was given and as a minimum I think they should outline (1) what their simulation process was, (2) what dose was administered (including Gy per fraction (which is to be fair given in the first para in procedures) and fractions per week) and broadly to what target volume (including the point to which the dose was prescribed), (3) which technique was used to deliver the radiation, (4) which particle was delivered (I assume

photon) and of what energy (I assume 6 or 10MV) and (5) whether there were any radiotherapy quality assurance procedures. I suspect most rad oncs would also want at least a mention of whether they used protocols relating to gastric emptying pre-Rx given that it's hard to review tox without this info. Also, if the RT wasn't standardised or there wasn't QA then this should be explicitly outlined as a limitation in the discussion.

Response: We appreciate your comments and the opportunity to provide additional details regarding the radiotherapy process used in our study. **Simulation Process:** Prior to radiotherapy, computed tomography (CT) simulation set as axial scanning with a layer thickness of 3 mm, complemented with abdominal compression, four-dimensional CT, and respiratory gating technologies, was performed to manage respiratory movement and accurately localize the target area. **Dose Administration and Target:** The gross tumor volume (GTV) included primary tumors and metastatic lymph nodes. The clinical target volume (CTV) included GTV and high-risk lymphatic drainage area. The planned target volume (PTV) included CTV with a 5-mm expansion in all directions. The radiotherapy treatment plan included a total dose of 45 Gy, typically administered in 25 fractions over five weeks. **Radiation Technique and Particle:** We used conformal, intensity-modulated, or spiral tomographic intensity-modulated techniques for delivering radiotherapy. The radiation was delivered using photons, typically at an energy level of 6 MV or 8 MV for conformal radiotherapy and 6 MV for intensity-modulated radiotherapy. **Quality Assurance Procedures:** Our study included rigorous quality assurance (QA) processes, managed by an experienced QA team from diverse fields.

The details of radiotherapy were added in the Methods as suggested, though due to the word limit of the manuscript, the protocol of radiotherapy might provide more informative details. Please kindly review.

Page 6/7: It seems to be suggested that they should not adjust for multiple significance testing so as to be 'hypothesis generating', whereas obviously they are just then finding things by chance instead. It looks from the tracks like they added in something about repeated measures corrections to the stats section and then deleted it again. The biomarker figure should only be reported with p-values derived from correction for multiple significance testing.

Response: We appreciate the reviewer's suggestion regarding the need for adjustments in the context of multiple significance testing. We concur with the perspective that such corrections help to minimize the risk of Type I errors. However, we would like to clarify the intention behind our preliminary biomarker analysis, which was of an exploratory nature. The goal of this analysis was to generate hypotheses for future targeted investigations rather than confirm specific hypotheses. In addition, Bonferroni correction test might be overly conservative and lead to irrelevant null hypotheses. Nevertheless, in response to your concern, we have performed corrections for multiple testing and provided the adjusted p-values in the supplementary table and in the main text: "After adjusting by Bonferroni correction test, the levels of CD56⁺ and CD56^{dim} cells in stroma were positively correlated with efficacy with an adjusted P-value of 0.024 and 0.012, respectively, while adjusted P-values of other biomarkers were all >0.05." We believe this step enhances the rigor of our results while maintaining the exploratory nature of the analysis.

Page 9: I think the fatigue data are so far from what would be expected that they need to highlight it in limitations and note that their tox reporting may well be underplaying the true toxicity.

Response: We appreciate the reviewer's comment. We have included this point in the limitations section of our paper. Here is added: Only one case of grade 2 fatigue was reported in our study, a finding which seems to diverge from expectations. The relatively small sample size may have led to an incomplete capture of patient experiences. Additionally, the potential for less stringent reporting and observation practices of adverse reactions such as fatigue could have been a contributing factor. Fatigue is inherently subjective, and thus its accurate assessment can be challenging.

Page 10: I think they should report in the manuscript itself that they used prophylactic antibiotics for those who had G4 leucopenia.

Response: We thank you for your helpful suggestion. Patients experiencing grade IV leukopenia with fever were administered prophylactic anti-infective therapy with broad-spectrum antibiotics. The antibiotics regimens were chosen based on experience and were adjusted according to the manifestation and results of culture. The antibiotics were discontinued once the leucopenia improved with no fever occurred. It was clarified in the methods.

Page 13: worth stating explicitly that the CROSS stats we asked them to reference refer to adenocarcinomas

Response: We thank you for your helpful suggestion. We agree and revised the manuscript to explicitly mention patients with adenocarcinomas.

Revised sentence in the manuscript: In the neoadjuvant setting, the CROSS trial reported a pCR rate of 23% specifically in patients diagnosed with esophagus or GEJ adenocarcinoma, while the POET trial reported a pCR rate of 15.6% in patients with esophagus or GEJ cancer [6, 7].

Reviewer #2

Thanks for the reply from the authors, most of the responses answered my concerns very well. However, the reason for choosing a pCR rate of 15% as a reference was not convincing. Tumor stage was a reason that led to a higher pCR rate in Ajani's studies, but the pCR rate was 16% in the FLOT arm of the FLOT4 trial. The key point is what the standard of care is that this study is being compared to. Please provide further information for this question, "These numbers were discussed and selected based on the differences in patient characteristics" was not a good answer.

Response: We appreciate the reviewer's insightful comments. We understand the concerns raised and would like to provide additional context to our decision-making process.

As the reviewer correctly pointed out, the pCR rates vary across studies due to differences in neoadjuvant therapies and patient populations. In the standard of care for G/GEJ cancer, the neoadjuvant therapy is primarily chemotherapy-based. Hence, we utilized the pCR rate from chemotherapy-based studies as our benchmark. In the RESOLVE study, a pCR rate of 5.6% was reported among a patient population with more advanced disease [8], while in the PRODIGY study, the rate was 10.4% [9], and in the FLOT4-AIO trial, a rate of 16% was achieved [10]. These rates reflect the variability in outcomes from chemotherapy as a neoadjuvant therapy in this patient population. Our choice of a 15% pCR rate as a reference point, thus, was a pragmatic decision that took into account these varied pCR rates from different chemotherapy studies. This was not just a purely statistical decision but also informed by our clinical experience treating this patient population.

Given these factors, we deemed it appropriate to select a 15% pCR rate for our null hypothesis. It represents an optimistic yet realistic benchmark grounded in the outcomes of standard neoadjuvant chemotherapy. We acknowledge that this value is a crucial parameter in our study, and we considered a wide range of factors, including previously reported data and clinical experience, to ensure it was both scientifically and clinically relevant. We trust that this provides a more comprehensive understanding of our rationale for this selection. It was clarified in the sample size calculation in the method: The null hypothesis was a pCR rate of 15% was determined based on our clinical experience and the pCR rate of 16% reported in the FLOT4-AIO trial,⁷⁹ as chemotherapy was the standard of care. The alternative hypothesis was a pCR rate of $\geq 35\%$ in this study.

Reviewer #3

The authors have addressed my concerns with this revision.

Response: My co-authors and I are very grateful to you for the time you have given in the appraisal of our paper and for the constructive feedback.

References

1. Tang, Z., et al., *The Neo-PLANET phase II trial of neoadjuvant camrelizumab plus concurrent chemoradiotherapy in locally advanced adenocarcinoma of stomach or gastroesophageal junction*. Nat Commun, 2022. **13**(1): p. 6807.
2. Li, S., et al., *Neoadjuvant therapy with immune checkpoint blockade, antiangiogenesis, and chemotherapy for locally advanced gastric cancer*. Nat Commun, 2023. **14**(1): p. 8.
3. Hasegawa, H., et al., *A multicenter, open-label, single-arm phase I trial of neoadjuvant nivolumab monotherapy for resectable gastric cancer*. Gastric Cancer : Official Journal of the International Gastric Cancer Association and the Japanese Gastric Cancer Association, 2022. **25**(3): p. 619-628.
4. Zhu, M., et al., *Pembrolizumab in Combination with Neoadjuvant Chemoradiotherapy for Patients with Resectable Adenocarcinoma of the Gastroesophageal Junction*. Clinical Cancer Research : an Official Journal of the American Association For Cancer Research, 2022. **28**(14): p. 3021-3031.
5. Tang, X., et al., *Neoadjuvant PD-1 blockade plus chemotherapy induces a high pathological complete response rate and anti-tumor immune subsets in clinical stage III gastric cancer*. Oncoimmunology, 2022. **11**(1): p. 2135819.
6. Shapiro, J., et al., *Neoadjuvant chemoradiotherapy plus surgery versus surgery alone for oesophageal or junctional cancer (CROSS): long-term results of a randomised controlled trial*. The Lancet Oncology, 2015. **16**(9): p. 1090-1098.
7. Stahl, M., et al., *Preoperative chemotherapy versus chemoradiotherapy in locally advanced adenocarcinomas of the oesophagogastric junction (POET): Long-term results of a controlled randomised trial*. Eur J Cancer, 2017. **81**: p. 183-190.
8. Zhang, X., et al., *Perioperative or postoperative adjuvant oxaliplatin with S-1 versus adjuvant oxaliplatin with capecitabine in patients with locally advanced gastric or gastro-oesophageal junction adenocarcinoma undergoing D2 gastrectomy (RESOLVE): an open-label, superiority and non-inferiority, phase 3 randomised controlled trial*. The Lancet. Oncology, 2021. **22**(8): p. 1081-1092.
9. Kang, Y.K., et al., *PRODIGY: A Phase III Study of Neoadjuvant Docetaxel, Oxaliplatin, and S-1 Plus Surgery and Adjuvant S-1 Versus Surgery and Adjuvant S-1 for Resectable Advanced Gastric Cancer*. J Clin Oncol, 2021. **39**(26): p. 2903-2913.
10. Al-Batran, S.E., et al., *Histopathological regression after neoadjuvant docetaxel, oxaliplatin, fluorouracil, and leucovorin versus epirubicin, cisplatin, and fluorouracil or capecitabine in patients with resectable gastric or gastro-oesophageal junction adenocarcinoma (FLOT4-AIO): results from the phase 2 part of a multicentre, open-label, randomised phase 2/3 trial*. Lancet Oncol, 2016. **17**(12): p. 1697-1708.

REVIEWERS' COMMENTS

Reviewer #1 (Remarks to the Author):

[none]

Reviewer #2 (Remarks to the Author):

All my concerns have been addressed, well done.

POINT-BY-POINT RESPONSES TO THE COMMENTS OF REVIEWER #1

Reviewer #1 (Remarks to the Author):

[none]

Response: We thank the Reviewer for the approve and all comments that improve the quality of this manuscript.

POINT-BY-POINT RESPONSES TO THE COMMENTS OF REVIEWER #2

Reviewer #2 (Remarks to the Author):

All my concerns have been addressed, well done.

Response: We thank the Reviewer for the recognition. The previously brought concerns were helpful for us to better report the findings of this study and were deeply appreciated.